# GRAPH GENERATION WITH $K^2$–TREES

**Yunhui Jang, Dongwoo Kim, Sungsoo Ahn**
Pohang University of Science and Technology
`{uni5510, dongwookim, sungsoo.ahn}@postech.ac.kr`

## ABSTRACT

Generating graphs from a target distribution is a significant challenge across many domains, including drug discovery and social network analysis. In this work, we introduce a novel graph generation method leveraging $K^2$–tree representation, originally designed for lossless graph compression. The $K^2$–tree representation encompasses inherent hierarchy while enabling compact graph generation. In addition, we make contributions by (1) presenting a sequential $K^2$–tree representation that incorporates pruning, flattening, and tokenization processes and (2) introducing a Transformer-based architecture designed to generate the sequence by incorporating a specialized tree positional encoding scheme. Finally, we extensively evaluate our algorithm on four general and two molecular graph datasets to confirm its superiority for graph generation.

## 1   INTRODUCTION

Generating graph-structured data is a challenging problem in numerous fields, such as molecular design (Li et al., 2018; Maziarka et al., 2020), social network analysis (Grover et al., 2019), and public health (Yu et al., 2020). Recently, deep generative models have demonstrated significant potential in addressing this challenge (Simonovsky & Komodakis, 2018; Jo et al., 2022; Vignac et al., 2022). In contrast to the classic random graph models (Albert & Barabási, 2002; Erdős et al., 1960), these methods leverage powerful deep generative paradigms, e.g., variational autoencoders (Simonovsky & Komodakis, 2018), normalizing flows (Madhawa et al., 2019), and diffusion models (Jo et al., 2022).

The graph generative models can be categorized into three types by the graph representation the models generate. First, an adjacency matrix is the most common representation (Simonovsky & Komodakis, 2018; Madhawa et al., 2019; Liu et al., 2021). Secondly, a string-based representation extracted from depth-first tree traversal on a graph can represent the graph as a sequence (Ahn et al., 2022; Goyal et al., 2020; Krenn et al., 2019). Finally, representing a graph as a composition of connected motifs, i.e., frequently appearing subgraphs, can preserve the high-level structural properties (Jin et al., 2018; 2020). We describe the representations on the left of Figure 1.

Although there is no consensus on the best graph representation, two factors drive their development. First is the need for compactness to reduce the complexity of graph generation and simplify the search space over graphs. For example, to generate a graph with $N$ vertices and $M$ edges, the adjacency matrix requires specifying $N^2$ elements. In contrast, the string representation typically requires specifying $O(N + M)$ elements, leveraging the graph sparsity (Ahn et al., 2022; Goyal et al., 2020; Segler et al., 2018). Motif representations also save space by representing frequently appearing subgraphs by basic building blocks (Jin et al., 2018; 2020).

The second factor driving the development of new graph representations is the presence of a hierarchy in graphs. For instance, community graphs possess underlying clusters, molecular graphs consist of distinct chemical fragments, and grid graphs exhibit a repetitive coarse-graining structure. In this context, motif representations (Jin et al., 2018; 2020) address the presence of a hierarchy in graphs; however, they are limited to a fixed vocabulary of motifs observed in the dataset or a specific domain.

**Contribution.** In this paper, we propose a novel graph generation framework, coined **H**ierarchical **G**raph **G**eneration with $K^2$–**T**ree (HGGT), which can represent not only non-attributed graphs but also attributed graphs in a compact and hierarchical way without domain-specific rules. The right-side table of Figure 1 emphasizes the benefits of HGGT. Since the $K^2$–tree recursively redefines

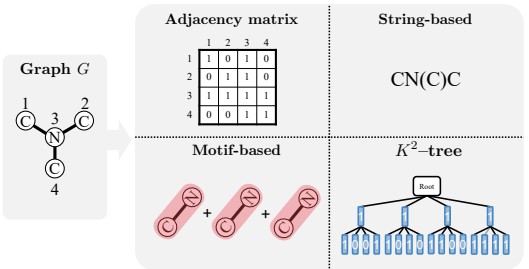

Figure 1: **(Left) Various representations used for graph generation. (Right) Comparing graph generative methods in terms of used graph representation.** The comparison is made with respect to a method being hierarchical (H), able to handle attributed graphs (A), and domain-agnostic (DA).

a graph into $K^2$ substructures, our representation becomes more compact and enables consideration of hierarchical structure in adjacency matrices.[1]

Specifically, we model the process of graph generation as an autoregressive construction of the $K^2$–tree. To this end, we design a sequential $K^2$–tree representation that recovers the original $K^2$–tree when combined sequentially. In particular, we propose a two-stage procedure where (1) we prune the $K^2$–tree to remove redundancy arising from the symmetric adjacency matrix for undirected graphs and (2) subsequently flatten and tokenize the $K^2$–tree into a sequence to minimize the number of decisions required for the graph generation.

We employ the Transformer architecture (Vaswani et al., 2017) to generate the sequential $K^2$–tree representation of a graph. To better incorporate the positional information of each node in a tree, we design a new positional encoding scheme specialized to the $K^2$–tree structure. Specifically, we represent the positional information of a node by its pathway from the root node; the proposed encoding enables the reconstruction of the full $K^2$–tree given just the positional information.

To validate the effectiveness of our algorithm, we test our method on popular graph generation benchmarks across six graph datasets: Community, Enzymes (Schomburg et al., 2004), Grid, Planar, ZINC (Irwin et al., 2012), and QM9 (Ramakrishnan et al., 2014). Our empirical results confirm that HGGT significantly outperformed existing graph generation methods on five out of six benchmarks, verifying the capability of our approach for high-quality graph generation across diverse applications.

To summarize, our key contributions are as follows:

- We propose a new graph generative model based on adopting the $K^2$–tree as a compact, hierarchical, and domain-agnostic representation of graphs.

- We introduce a novel, compact sequential $K^2$–tree representation obtained from pruning, flattening, and tokenizing the $K^2$–tree.

- We propose an autoregressive model to generate the sequential $K^2$–tree representation using Transformer architecture with a specialized positional encoding scheme.

- We validate the efficacy of our framework by demonstrating state-of-the-art graph generation performance on five out of six graph generation benchmarks.

## 2  RELATED WORK

**Graph representations for graph generation.** The choice of graph representation is a crucial aspect of graph generation, as it significantly impacts the efficiency and allows faithful learning of the generative model. The most widely used one is the adjacency matrix, which simply encodes the pairwise relationship between nodes (Jo et al., 2022; Vignac et al., 2022; You et al., 2018; Liao et al., 2019; Shi et al., 2020; Luo et al., 2021; Kong et al., 2023; Chen et al., 2023). However, several methods (Vignac et al., 2022; You et al., 2018; Jo et al., 2022) suffer from the high complexity in generating the adjacency matrix, especially for large graphs.

---

[1]This differs from the conventional hierarchical community structure. We provide the discussion in Appendix H.

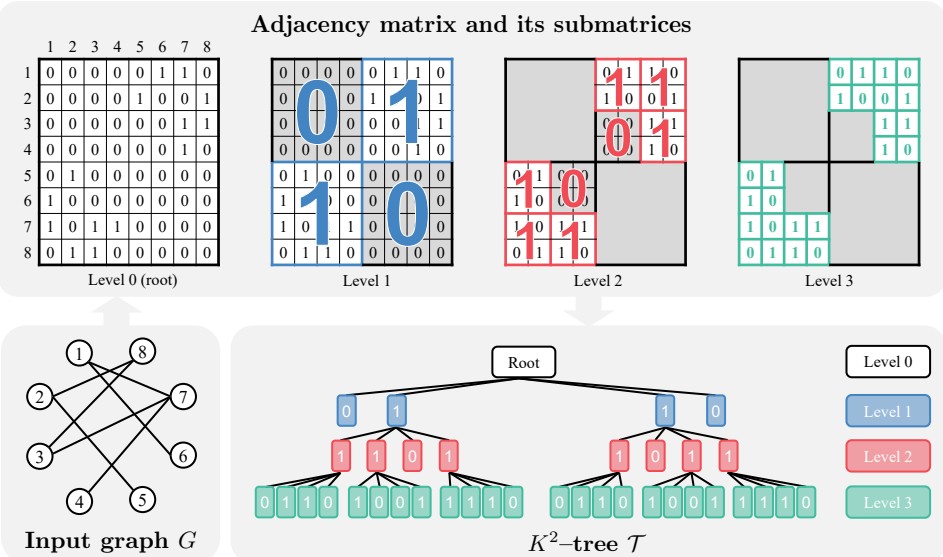

Figure 2: $K^2$–**tree with** $K = 2$. The $K^2$–tree describes the hierarchy of the adjacency matrix iteratively being partitioned to $K \times K$ submatrices. It is compact due to summarizing any zero-filled submatrix with a size larger than $1 \times 1$ (shaded in grey) by a leaf node $u$ with label $x_u = 0$.

To address this issue, researchers have developed graph generative models that employ alternative graph representations such as motif-based representations and string-based representations. For instance, Ahn et al. (2022); Segler et al. (2018) proposed to generate molecule-specific string representations, and Jin et al. (2018; 2020); Yang et al. (2021) suggested generative models that extract reasonable fragments from data and generate the set of motifs. However, these methods rely on domain-specific knowledge and are restricted to molecular data.

**Lossless graph compression.** Lossless graph compression (Besta & Hoefler, 2018) aims to reduce the size and complexity of graphs while preserving their underlying structures. Specifically, several works (Brisaboa et al., 2009; Raghavan & Garcia-Molina, 2003) introduced hierarchical graph compression methods that compress graphs leveraging their hierarchical structure. In addition, Bouritsas et al. (2021) derived the compressed representation using a learning-based objective.

## 3   $K^2$–TREE REPRESENTATION OF A GRAPH

In this section, we introduce the $K^2$–tree as a hierarchical and compact representation of graphs, as originally proposed for graph compression (Brisaboa et al., 2009). In essence, the $K^2$–tree is a $K^2$-ary ordered tree that recursively partitions the adjacency matrix into $K \times K$ submatrices.[2] Its key idea is to summarize the submatrices filled only with zeros with a single tree-node, exploiting the sparsity of the adjacency matrix. From now on, we indicate the tree-node as a node. The representation is hierarchical, as it associates each parent and child node pair with a matrix and its corresponding submatrix, respectively, as described in Figure 2.

To be specific, we consider the $K^2$–tree representation $(\mathcal{T}, \mathcal{X})$ of an adjacency matrix $A$ as a $K^2$-ary tree $\mathcal{T} = (\mathcal{V}, \mathcal{E})$ associated with binary node attributes $\mathcal{X} = \{x_u : u \in \mathcal{V}\}$. Every non-root node is uniquely indexed as $(i, j)$-th child of its parent node for some $i, j \in \{1, \dots, K\}$. The tree $\mathcal{T}$ is ordered so that every $(i, j)$-th child node is ranked $K(i-1) + j$ among its siblings. Then the $K^2$–tree satisfies the following conditions:

- Each node $u$ is associated with a submatrix $A^{(u)}$ of the adjacency matrix $A$.

- If the submatrix $A^{(u)}$ for a node $u$ is filled only with zeros, $x_u = 0$. Otherwise, $x_u = 1$.

- A node $u$ is a leaf node if and only if $x_u = 0$ or the matrix $A^{(u)}$ is a $1 \times 1$ matrix.

---

[2]By default, we assume the number of nodes in the original graph to be the power of $K^2$.

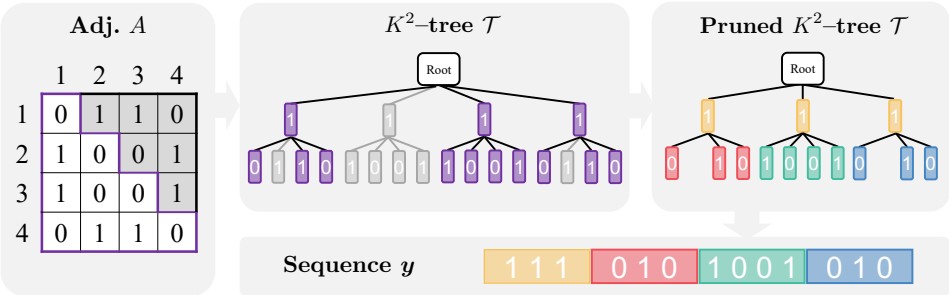

Figure 3: **Illustration of the sequential representation for $K^2$–tree.** The shaded parts of the adjacency matrix $A$ and the $K^2$–tree $\mathcal{T}$ denote redundant parts, which are further pruned, while the purple-colored parts of $A$ and $\mathcal{T}$ denote non-redundant parts. Also, same-colored tree-nodes of pruned $K^2$–tree are grouped and tokenized into the same colored parts of the sequence $\boldsymbol{y}$.

- Let $B_{1,1}, \ldots, B_{K,K}$ denote the $K \times K$ partitioning of the matrix $A^{(u)}$ with $i, j$ corresponding to row- and column-wise order, respectively. The child nodes $v_{1,1}, \ldots, v_{K,K}$ of the tree-node $u$ are associated with the submatrices $B_{1,1}, \ldots, B_{K,K}$, respectively.

The generated $K^2$–tree is a compact description of graph $G$ as any node $u$ with $x_u = 0$ and $d_u < \max_u d_u$ where $d_u$ is the distance from the root. summarizes a large submatrix filled only with zeros. In the worst-case scenario, the size of the $K^2$–tree is $MK^2(\log_{K^2}(N^2/M) + O(1))$ (Brisaboa et al., 2009), where $N$ and $M$ denote the number of nodes and edges in the original graph, respectively. This constitutes a significant improvement over the $N^2$ size of the full adjacency matrix.

Additionally, the $K^2$–tree is hierarchical ensuring that (1) each tree node represents the connectivity between a specific set of nodes, and (2) nodes closer to the root correspond to a larger set of nodes. We emphasize that the nodes associated with submatrices overlapping with the diagonal of the original adjacency matrix indicate intra-connectivity within a group of nodes. In contrast, the remaining nodes describe the interconnectivity between two distinct sets of nodes.

We also describe the detailed algorithms for constructing a $K^2$–tree from a given graph $G$ and recovering a graph from the $K^2$–tree in Appendices A and B, respectively. It is crucial to note that the ordering of the nodes in the adjacency matrix influences the $K^2$–tree structure. Inspired by Diamant et al. (2023), we adopt Cuthill-McKee (C-M) ordering as our ordering scheme. We empirically discover that C-M ordering (Cuthill & McKee, 1969) provides the most compact $K^2$–tree.[3] Our explanation is that the C-M ordering is specifically designed to align the non-zero elements of a matrix near its diagonal so that there is a higher chance of encountering large submatrices filled only with zeros, which can be efficiently summarized in the $K^2$–tree representation.

## 4  HIERARCHICAL GRAPH GENERATION WITH $K^2$–TREES

In this section, we present our novel method, hierarchical graph generation with $K^2$–trees (HGGT), exploiting the hierarchical and compact structure of the $K^2$–tree representation of a graph. In detail, we transform the $K^2$–tree into a highly compressed sequence through a process involving pruning and tokenization. Subsequently, we employ a Transformer enhanced with tree-based positional encodings, for the autoregressive generation of this compressed sequence.

### 4.1  SEQUENTIAL $K^2$–TREE REPRESENTATION

Here, we propose an algorithm to flatten the $K^2$–tree into a sequence, which is essential for the autoregressive generation of the $K^2$–tree. In particular, we aim to design a sequential representation that is even more compact than the $K^2$–tree to minimize the number of decisions required for the generation of the $K^2$–tree. To this end, we propose (1) pruning $K^2$–tree by removing redundant nodes, (2) flattening the pruned $K^2$–tree into a sequence, and (3) applying tokenization based on the $K^2$–tree structure. We provide an illustration of the overall process in Figure 3.

---

[3]We provide the results in Section 5.3.

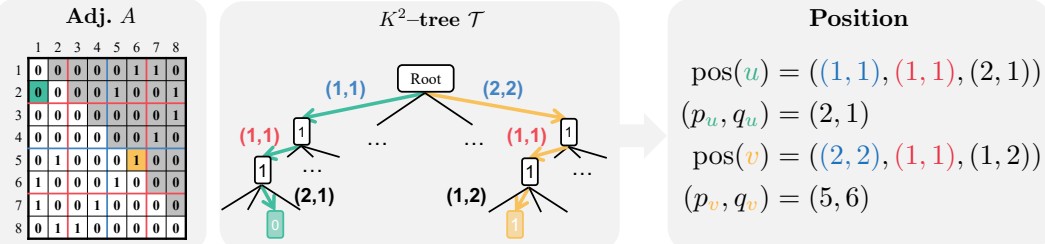

Figure 4: **Illustration of the tree-node positions of $K^2$–tree.** The shaded parts of the adjacency matrix denote redundant parts, e.g., $p_u < q_u$. Additionally, colored elements correspond to tree-nodes of the same color and the same-colored tree-edges signify the root-to-target downward path. Blue and red tuples denote the order in the first and second levels, respectively. The tree node $u$ is non-redundant as $p_u > q_u$ while $v$ is redundant as $p_v < q_v$.

**Pruning the $K^2$–tree.** To obtain the pruned $K^2$–tree, we identify and eliminate redundant nodes due to the symmetry of the adjacency matrix for undirected graphs. In particular, without loss of generality, such nodes are associated with submatrices positioned above the diagonal since they mirror the counterparts located below the diagonal.

To this end, we now describe a formula to identify redundant nodes based on the position of a submatrix $A^{(u)}$, tied to a specific node $u$ at depth $L$, within the adjacency matrix $A$. Let $v_0, v_1, \dots, v_L$ be a sequence of nodes representing a downward path from the root node $r = v_0$ to the node $u = v_L$. With $(i_{v_\ell}, j_{v_\ell})$ denoting the order of $v_\ell$ among its $K \times K$ siblings, the node position can be represented as $\text{pos}(u) = ((i_{v_1}, j_{v_1}), \dots, (i_{v_L}, j_{v_L}))$. Note that node $u$ at depth $L$ corresponds to an element of $K^L \times K^L$ partitions of the adjacency matrix $A$. The row and column indexes of the submatrix $A^{(u)}$ are derived as the $(p_u, q_u) = (\sum_{\ell=1}^{L} K^{L-\ell}(i_{v_\ell} - 1) + 1, \sum_{\ell=1}^{L} K^{L-\ell}(j_{v_\ell} - 1) + 1)$ as illustrated in Figure 4. As a result, we eliminate any node associated with a submatrix above the diagonal, i.e., we remove node $u$ when $p_u < q_u$.

Consequently, the pruned $K^2$–tree maintains only the nodes associated with submatrices devoid of redundant nodes, i.e., those containing elements of the adjacency matrix positioned at the diagonal or below the diagonal. Notably, following this pruning process, the $K^2$–tree no longer adheres to the structure of a $K \times K$-ary tree. Additionally, consider a non-leaf node $u$ is associated with a submatrix $A^{(u)}$ that includes any diagonal elements of the adjacency matrix $A$. Then the node $u$ possess $K(K + 1)/2$ child nodes after pruning $K(K - 1)/2$ child nodes associated with the redundant submatrices. Otherwise, the non-leaf node $u$ remains associated with $K \times K$ child nodes. Note that our framework can be extended to directed graphs by omitting the pruning process.

**Flattening and tokenization of the pruned $K^2$–tree.** Next, we explain how to obtain a sequential representation of the pruned $K^2$–tree based on flattening and tokenization. Our idea is to flatten a $K^2$–tree as a sequence of node attributes $\{x_u : u \in \mathcal{V}\}$ using breadth-first traversal and then to tokenize the sequence by grouping the nodes that share the same parent node, i.e., sibling nodes.

For this purpose, we denote the sequence of nodes obtained from a breadth-first traversal of non-root nodes in the $K^2$–tree as $u_1, \dots, u_{|\mathcal{V}|-1}$, and the corresponding sequence of node attributes as $\boldsymbol{x} = (x_1, \dots, x_{|\mathcal{V}|-1})$. It is important to note that sibling nodes sharing the same parent appear sequentially in the breadth-first traversal.

Next, by grouping the sibling nodes, we tokenize the sequence $\boldsymbol{x}$. As a result, we obtain a sequence $\boldsymbol{y} = (y_1, \dots, y_T)$ where each element is a token representing a group of attributes associated with sibling nodes. For example, the $t$-th token corresponding to a group of $K^2$ sibling nodes is represented by $y_t = (x_{v_{1,1}}, \dots, x_{v_{K,K}})$ where $v_{1,1}, \dots, v_{K,K}$ share the same parent node $u$. Such tokenization allows representing the whole $K^2$–tree using $M(\log_{K^2}(N^2/M) + O(1))$ space, where $N$ and $M$ denote the number of nodes and edges in the original graph, respectively.

We highlight that the number of elements in each token $y_t$ may vary due to the pruned $K^2$–tree no longer being a $K \times K$-ary tree, as mentioned above. With this in consideration, we generate a vocabulary of $2^{K^2} + 2^{K(K+1)/2}$ potential configurations for each token $y_t$. This vocabulary size is small in practice since we set the value $K$ to be small, e.g., setting $K = 2$ induces the size of 24.

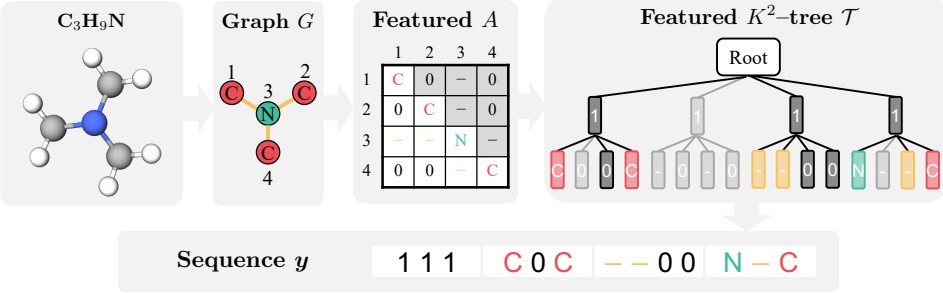

Figure 5: **An example of featured $K^2$–tree representation.** The shaded parts of the adjacency matrix and $K^2$–tree denote the redundant parts. The black-colored tree-nodes denote the normal tree-nodes with binary attributes while other-colored feature elements in the adjacency matrix $A$ denote the same-colored featured tree-nodes and sequence elements. The node features (i.e., C and N) and edge feature (i.e., single bond $-$) of the molecule are represented within the leaf nodes.

In particular, we remark that a token with $K(K+1)/2$ elements carries different semantics from another token with $K^2$ elements. The former corresponds to a submatrix situated on the adjacency matrix's diagonal, thus indicating connectivity *within* a set of nodes. In contrast, the latter relates to a submatrix illustrating connectivity *between* pairs of node sets. This supports our decision to assign distinct values to a token with $K(K+1)/2$ elements and another with $K^2$ elements, even when the tokens might represent the same combination of node features in the unpruned tree.

**Generating featured graphs.** We also extend our HGGT to graphs with node and edge-wise features, e.g., molecular graphs. At a high level, we apply our algorithm to the featured adjacency matrix, where each diagonal element corresponds to a node feature and each non-diagonal element corresponds to an edge feature. node attributes of leaf nodes in $K^2$–tree correspond to node and edge features, while attributes of non-leaf nodes are the same with the non-attributed $K^2$–trees (i.e., ones and zeros). See Figure 5 for an illustration and Appendix C for a complete description.

## 4.2 GENERATING $K^2$–TREE WITH TRANSFORMER AND $K^2$–TREE POSITIONAL ENCODING

We describe our algorithm to generate the sequence of $K^2$–tree representation $\boldsymbol{y} = (y_1, \ldots, y_T)$. We utilize the masked Transformer (Vaswani et al., 2017) to make predictions on $p_\theta(y_t|y_{t-1}, \ldots, y_1)$. To improve the model's understanding of the tree structure, we devise a tree-positional encoding. We also offer an algorithm to construct the $K^2$–tree from the sequence generated by the Transformer.

**Transformer with $K^2$–tree positional encoding.** We first introduce the Transformer architecture to parameterize the distribution $p_\theta(y_t|y_{t-1}, \ldots, y_1)$ for autorgressive generation. Briefly, the model is trained with self-attention, and during inference, it generates the sequence one token at a time, relying on the previously generated sequence. To account for tree structural information, we incorporate tree-positional encodings for each time-step $t$.

During training, we mask the attention layer to ensure that predictions at each step are not influenced by future tokens of the sequence. The objective function is maximum likelihood, denoted by $\max \log p(\boldsymbol{y})$, where $p(\boldsymbol{y}) = p(y_1)\Pi_{t=2}^{T}p(y_t|y_{1:t-1})$. This objective aims to maximize the probability of predicting the next token correctly based on the preceding tokens.

For inference, we begin the process with a begin-of-sequence (BOS) token as the input to our trained Transformer decoder. The model then computes the distribution of potential tokens for the next step, denoted by $p(y_t|y_{1:t-1})$, and the next token is sampled from this distribution. This token is appended to the input sequence, and the extended sequence is fed back into the model to generate the subsequent token. This iterative procedure is terminated when a predefined maximum length is reached or an end-of-sequence (EOS) token emerges.

To enhance the input $y_t$, we incorporate the positional encoding for $u$. As outlined in Section 4.1, the node attributes in $y_t$ are associated with child nodes of a particular node $u$. Therefore, the encoding is based on the downward path from the root node $r = v_0$ to the node $u = v_L$, represented as $(v_0, \ldots, v_L)$. In this context, the order of $v_\ell$ amongst its siblings in the non-pruned $K^2$–tree is denoted as a tuple $(i_{v_\ell}, j_{v_\ell})$. Subsequently, we further update the input feature $y_t$ with positional

Table 1: **Generic graph generation performance.** The baseline results are from prior works (Jo et al., 2022; Liao et al., 2019; Martinkus et al., 2022; Luo et al., 2022) or public codes (marked by *). For each metric, the best number is highlighted in **bold** and the second-best number is underlined.

| Method | Community-small $12 \leq |V| \leq 20$ | | | Planar $|V| = 64$ | | | Enzymes $10 \leq |V| \leq 125$ | | | Grid $100 \leq |V| \leq 400$ | | |
|---|---|---|---|---|---|---|---|---|---|---|---|---|
| | Deg. | Clus. | Orb. | Deg. | Clus. | Orb. | Deg. | Clus. | Orb. | Deg. | Clus. | Orb. |
| GraphVAE | 0.350 | 0.980 | 0.540 | - | - | - | 1.369 | 0.629 | 0.191 | 1.619 | **0.000** | 0.919 |
| GraphRNN | 0.080 | 0.120 | 0.040 | 0.005 | 0.278 | 1.254 | 0.017 | 0.062 | 0.046 | 0.064 | 0.043 | 0.021 |
| GNF | 0.200 | 0.200 | 0.110 | - | - | - | - | - | - | - | - | - |
| GRAN* | 0.005 | 0.142 | 0.090 | 0.001 | 0.043 | 0.001 | 0.023 | 0.031 | 0.169 | 0.001 | 0.004 | 0.002 |
| EDP-GNN | 0.053 | 0.144 | 0.026 | - | - | - | 0.023 | 0.268 | 0.082 | 0.455 | 0.238 | 0.328 |
| GraphGen* | 0.075 | 0.065 | 0.014 | 1.762 | 1.423 | 1.640 | 0.146 | 0.079 | 0.054 | 1.550 | 0.017 | 0.860 |
| GraphAF | 0.180 | 0.200 | 0.020 | - | - | - | 1.669 | 1.283 | 0.266 | - | - | - |
| GraphDF | 0.060 | 0.120 | 0.030 | - | - | - | 1.503 | 1.283 | 0.266 | - | - | - |
| SPECTRE | - | - | - | 0.010 | 0.067 | 0.010 | - | - | - | - | - | - |
| GDSS | 0.045 | 0.086 | 0.007 | 0.250 | 0.393 | 0.587 | 0.026 | 0.061 | 0.009 | 0.111 | 0.005 | 0.070 |
| DiGress* | 0.012 | 0.025 | 0.002 | **0.000** | 0.002 | 0.008 | 0.011 | 0.039 | 0.010 | 0.016 | **0.000** | 0.004 |
| GDSM | 0.011 | 0.015 | **0.001** | - | - | - | 0.013 | 0.088 | 0.010 | 0.002 | **0.000** | **0.000** |
| HGGT (ours) | **0.001** | **0.006** | 0.003 | **0.000** | **0.001** | **0.000** | **0.005** | **0.017** | **0.000** | **0.000** | **0.000** | **0.000** |

| (a) Train | (b) GraphGen | (c) GDSS | (d) DiGress | (e) HGGT (ours) |

Figure 6: **Generated samples for Community-small (top), and Grid (bottom) datasets.**

encoding, which is represented as $\mathrm{PE}(u) = \sum_{\ell=1}^{L} \phi_\ell(i_{v_\ell}, j_{v_\ell})$, where $\phi$ denotes the embedding function that converts the order tuple into vector representations and $((i_{v_1}, j_{v_1}), \ldots, (i_{v_L}, j_{v_L}))$ is the sequence of orders of a downward path from $r$ to $u$.

**Constructing $K^2$–tree from the sequential representation.** We next explain the algorithm to recover a $K^2$–tree from its sequential representation $\boldsymbol{y}$. In particular, we generate the $K^2$–tree simultaneously with the sequence to incorporate the tree information for each step of the autoregressive generation. The algorithm begins with an empty tree containing only a root node and iteratively expands each "frontier" node based on the sequence of the decisions made by the generative model. To facilitate a breadth-first expansion approach, the algorithm utilizes a first-in-first-out (FIFO) queue, which contains node candidates to be expanded.

To be specific, our algorithm initializes a $K^2$–tree $\mathcal{T} = (\{r\}, \emptyset)$ with the root node $r$ associated with the node attribute $x_r = 1$. It also initializes the FIFO queue $\mathcal{Q}$ with $r$. Then at each $t$-th step, our algorithm expands the node $u$ popped from the queue $\mathcal{Q}$ using the token $y_t$. To be specific, for each node attribute $x$ in $y_t$, our algorithm adds a child node $v$ with $x_v = x$. If $x = 1$ and the size of $A^{(v)}$ is larger than $1 \times 1$, the child node $v$ is inserted into the queue $Q$. This algorithm is designed to retrieve the pruned tree, which allows the computation of positional data derived from the $y_t$ information.

## 5 EXPERIMENT

### 5.1 GENERIC GRAPH GENERATION

**Experimental setup.** We first validate the general graph generation performance of our HGGT on four popular graph benchmarks: (1) **Community-small**, 100 community graphs, (2) **Planar**, 200

Table 2: **Molecular graph generation performance.** The baseline results are from prior works (Jo et al., 2022; Luo et al., 2022) or obtained by running the open-source codes (denoted by *). The best results are highlighted in **bold** and the second best results are underlined.

| Method | QM9 | | | | | ZINC250k | | | | |
|---|---|---|---|---|---|---|---|---|---|---|
| | Val. ↑ | NSPDK ↓ | FCD ↓ | Uniq. ↑ | Nov. ↑ | Val. ↑ | NSPDK ↓ | FCD ↓ | Uniq. ↑ | Nov. ↑ |
| EDP-GNN | 47.52 | 0.005 | 2.68 | **99.25** | 86.58 | 82.97 | 0.049 | 16.74 | 99.79 | **100** |
| MoFlow | 91.36 | 0.017 | 4.47 | 98.65 | 94.72 | 63.11 | 0.046 | 20.93 | **99.99** | **100** |
| GraphAF | 74.43 | 0.020 | 5.27 | 88.64 | 86.59 | 68.47 | 0.044 | 16.02 | 98.64 | **100** |
| GraphDF | 93.88 | 0.064 | 10.93 | 98.58 | **98.54** | 90.61 | 0.177 | 33.55 | 99.63 | **100** |
| GraphEBM | 8.22 | 0.030 | 6.14 | 97.90 | 97.01 | 5.29 | 0.212 | 35.47 | 98.79 | **100** |
| GDSS | 95.72 | 0.003 | 2.9 | 98.46 | 86.27 | 97.01 | 0.019 | 14.66 | 99.64 | **100** |
| DiGress* | 99.01 | 0.001 | **0.25** | 96.34 | 35.46 | **100** | 0.042 | 16.54 | 99.97 | **100** |
| GDSM | **99.90** | 0.003 | 2.65 | - | - | 92.70 | 0.017 | 12.96 | - | - |
| HGGT (ours) | 99.22 | **0.000** | 0.40 | 95.65 | 24.01 | 92.87 | **0.001** | **1.93** | 99.97 | 99.83 |

planar graphs, (3) **Enzymes** (Schomburg et al., 2004), 587 protein tertiary structure graphs, and (4) **Grid**, 100 2D grid graphs. Following baselines, we adopt maximum mean discrepancy (MMD) to compare three graph property distributions between generated graphs and test graphs: degree (**Deg.**), clustering coefficient (**Clus.**), and 4-node-orbit counts (**Orb.**). We conduct all the experiments using a single RTX 3090 GPU. The detailed descriptions of our experimental setup are in Appendix D.

**Baselines.** We compare our HGGT with twelve graph generative models: GraphVAE (Simonovsky & Komodakis, 2018), GraphRNN (You et al., 2018), GNF Liu et al. (2019), GRAN (Liao et al., 2019), EDP-GNN (Niu et al., 2020), GraphGen (Goyal et al., 2020), GraphAF (Shi et al., 2020), GraphDF (Luo et al., 2021), SPECTRE (Martinkus et al., 2022), GDSS (Jo et al., 2022), DiGress (Vignac et al., 2022), and GDSM (Luo et al., 2022). A detailed implementation description is in Appendix E.

**Results.** Table 1 shows the experimental results. We observe that HGGT outperforms all baselines on all datasets. Note that our model consistently outperforms all baselines regardless of the graph sizes, indicating better generalization performance across various environments. In particular, we observe how the performance of HGGT is extraordinary for Grid. We hypothesize that HGGT better captures the hierarchical structure and repetitive local connectivity of the grid graphs than the other baselines. We also provide visualizations of the generated graphs in Figure 6.

## 5.2 MOLECULAR GRAPH GENERATION

**Experimental setup.** To test the ability of HGGT on featured graphs, we further conduct an evaluation of molecule generation tasks. We use two molecular datasets: QM9 (Ramakrishnan et al., 2014) and ZINC250k (Irwin et al., 2012). Following the previous work (Jo et al., 2022), we evaluate 10,000 generated molecules using five metrics: (a) validity (**Val.**), (b) neighborhood subgraph pairwise distance kernel (**NSPDK**), (c) Frechet ChemNet Distance (**FCD**), (d) uniqueness (**Uniq.**), and (e) novelty (**Nov.**). Note that NSPDK and FCD are measured between the generated samples and the test set. The validity, uniqueness, and novelty metrics are measured within the generated samples.

**Baselines.** We compare HGGT with eight deep graph generative models: EDP-GNN (Niu et al., 2020), MoFlow (Zang & Wang, 2020), GraphAF (Shi et al., 2020), GraphDF (Luo et al., 2021), GraphEBM (Liu et al., 2021), GDSS (Jo et al., 2022), DiGress (Vignac et al., 2022), and GDSM(Luo et al., 2022). We provide a detailed implementation description in Appendix E.

**Results.** The experimental results are reported in Table 2. We observe that HGGT showed competitive results on all the baselines on most of the metrics. The results suggest that the model can generate chemically valid features, i.e., atom types, accordingly, along with the structure of the graphs. In particular, for the ZINC250k dataset, we observe a large gap between our method and the baselines in NSPDK and FCD scores while showing competitive performance in the other metrics. Since FCD and NSPDK measure the similarity between molecular features and subgraph structures, respectively, HGGT can generate similar features and subgraphs observed in the real molecules.

## 5.3 ABLATION STUDIES

**Time complexity.** We conduct experiments to measure the inference time of the proposed algorithm. The results are presented in the upper left table of Figure 7, where we report the time to generate

| Time (sec) | Comm. | Planar | Enzymes | Grid |
|---|---|---|---|---|
| GRAN | 3.51 | 5.40 | 3.99 | 14.68 |
| GDSS | 0.54 | 8.85 | 1.09 | 25.90 |
| DiGress | 0.34 | 3.29 | 1.29 | 45.41 |
| HGGT (ours) | **0.03** | **0.58** | **0.09** | **8.16** |

| Method | Comm. | Planar | Enzymes | Grid |
|---|---|---|---|---|
| BFS | 0.534 | 0.201 | 0.432 | 0.048 |
| DFS | 0.619 | 0.204 | 0.523 | 0.064 |
| C-M | **0.508** | **0.195** | **0.404** | **0.045** |

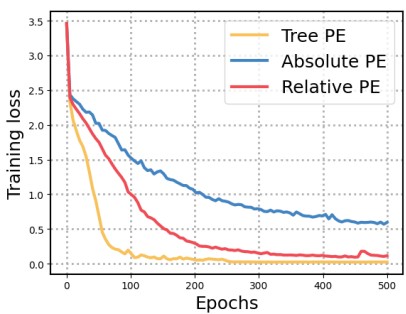

Figure 7: **(Upper left) Inference time to generate a single graph. (Lower left) Average compression ratio on various node orderings. (Right) Training loss on different positional encodings.**

Table 3: **Ablation study for algorithmic components of HGGT.**

| | | | Community-small | | | Planar | | | Enzymes | | |
|---|---|---|---|---|---|---|---|---|---|---|---|
| Group | TPE | Prune | Degree | Cluster. | Orbit | Degree | Cluster. | Orbit | Degree | Cluster. | Orbit |
| ✗ | ✗ | ✗ | 0.072 | 0.199 | 0.080 | 0.346 | 1.824 | 1.403 | 0.050 | 0.060 | 0.021 |
| ✓ | ✗ | ✗ | 0.009 | 0.105 | **0.001** | 0.003 | **0.001** | 0.002 | 0.005 | 0.022 | 0.007 |
| ✓ | ✓ | ✗ | 0.002 | 0.028 | **0.001** | 0.003 | **0.001** | 0.002 | **0.002** | 0.020 | 0.002 |
| ✓ | ✓ | ✓ | **0.001** | **0.006** | 0.003 | **0.000** | **0.001** | **0.000** | 0.005 | **0.017** | **0.000** |

a single sample. We can observe that HGGT generates a graph faster than the others due to the simplified representation.

**Adjacency matrix orderings.** It is clear that the choice of node ordering influences the size of $K^2$–tree. We validate our choice of Cuthill-McKee (C-M) ordering (Cuthill & McKee, 1969) by comparing its compression ratio to other node orderings: breadth-first search (BFS) and depth-first search (DFS). The compression ratio is defined as the number of elements in $K^2$–tree divided by $N^2$. In the left below table of Figure 7, we present the compression ratios for each node ordering. One can observe that C-M ordering shows the best ratio in all the datasets compared to others.

**Positional encoding.** In this experiment, we assess the impact of various positional encodings in our method. We compare our tree positional encoding (TPE) to absolute positional encoding (APE) (Vaswani et al., 2017) and relative positional encoding (RPE) (Shaw et al., 2018) on the Planar dataset. Our findings, as presented in the right figure of Figure 7, demonstrate that TPE outperforms other positional encodings with faster convergence of training loss. These observations highlight the importance of appropriate positional encoding for generating high-quality graphs.

**Ablation of algorithmic components.** We introduce three components to enhance the performance of HGGT: grouping into tokens (Group), incorporating tree positional encoding (TPE), and pruning the $K^2$–tree (Prune). To verify the effectiveness of each component, we present the experimental results for our method with incremental inclusion of these components. The experimental results are reported in Table 3. The results demonstrate the importance of each component in improving graph generation performance, with grouping being particularly crucial, thereby validating the significance of our additional components to the sequential $K^2$–tree representation.

## 6 CONCLUSION

In this paper, we presented a novel $K^2$–tree-based graph generative model (HGGT) which enables a compact, hierarchical, and domain-agnostic generation. Our experimental evaluation demonstrated state-of-the-art performance across various graph datasets. An interesting avenue for future work is the broader examination of other graph representations to graph generation, e.g., a plethora of representations (Boldi et al., 2009; Larsson & Moffat, 2000).

**Reproducibility** All experimental code related to this paper is available at `https://github.com/yunhuijang/HGGT`. Detailed insights regarding the experiments, encompassing dataset and model specifics, are available in Section 5. For intricate details like hyperparameter search, consult Appendix D. In addition, the reproduced dataset for each baseline is in Appendix E.

**Acknowledgements** This work partly was supported by Institute of Information & communications Technology Planning & Evaluation (IITP) grant funded by the Korea government (MSIT) (No. IITP-2019-0-01906, Artificial Intelligence Graduate School Program (POSTECH)), the National Research Foundation of Korea (NRF) grant funded by the Korea government (MSIT) (No. 2022R1C1C1013366), Basic Science Research Program through the National Research Foundation of Korea (NRF) funded by the Ministry of Education (2022R1A6A1A0305295413, 2021R1C1C1011375), and the Technology Innovation Program (No. 20014926, Development of BIT Convergent AI Architecture, Its Validation and Candidate Selection for COVID19 Antibody, Repositioning and Novel Synthetic Chemical Therapeutics) funded by the Ministry of Trans, Industry & Energy (MOTIE, Korea).

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

# A  CONSTRUCTION OF A $K^2$–TREE FROM THE GRAPH

---

**Algorithm 1** $K^2$–tree construction

---

    **Input**:Adjacency matrix $A$ and partitioning factor $K$.
1: Initialize the tree $\mathcal{T} \leftarrow (\mathcal{V}, \mathcal{E})$ with $\mathcal{V} = \emptyset, \mathcal{E} = \emptyset$.          $\triangleright$ $K^2$–tree.
2: Initialize an empty queue $Q$.          $\triangleright$ Candidates to be expanded into child nodes.
3: Set $\mathcal{V} \leftarrow \mathcal{V} \cup \{r\}$, $x_r \leftarrow 1$ and let $A^{(r)} \leftarrow A$. Insert $r$ into the queue $Q$.      $\triangleright$ Add root node $r$.
4: **while** $Q \neq \emptyset$ **do**
5:      Pop $u$ from $\mathcal{Q}$.
6:      **if** $x_u = 0$ **then**          $\triangleright$ Condition for not expanding the node $u$.
7:          Go to line 4.
8:      **end if**
9:      Update $s \leftarrow \dim(A^{(u)})/K$
10:      **for** $i = 1, \ldots, K$ **do**          $\triangleright$ Row-wise indices.
11:          **for** $j = 1, \ldots, K$ **do**          $\triangleright$ Column-wise indices.
12:              Set $B_{i,j} \leftarrow A^{(u)}[(i-1)s : is, (j-1)s : js]$.
                     $\triangleright$ Operation to obtain $s \times s$ submatrix $B_{i,j}$ of $A^{(u)}$.
13:             If $B_{i,j}$ is filled with zeros, set $x_v \leftarrow 1$. Otherwise, set $x_v \leftarrow 0$.
                     $\triangleright$ Update tree-node attribute.
14:             If $\dim(v_{i,j}) > 1$, update $\mathcal{Q} \leftarrow v_{i,j}$.
15:          **end for**
16:      **end for**
17:      Set $\mathcal{V} \leftarrow \mathcal{V} \cup \{v_{1,1}, \ldots, v_{K,K}\}$.          $\triangleright$ Update tree nodes.
18:      Set $\mathcal{E} \leftarrow \mathcal{E} \cup \{(u, v_{1,1}), \ldots, (u, v_{K,K})\}$.          $\triangleright$ Update tree edges.
19: **end while**
    **Output**: $K^2$–tree $(\mathcal{T}, \mathcal{X})$ where $\mathcal{X} = \{x_u : u \in \mathcal{V}\}$.

---

In this section, we explain our algorithm to construct a $K^2$–tree $(\mathcal{T}, \mathcal{X})$ from a given graph $G = A$ where $G$ is a symmetric non-featured graph and $A$ is an adjacency matrix. Note that he $K^2$-ary tree $\mathcal{T} = (\mathcal{V}, \mathcal{E})$ is associated with binary node attributes $\mathcal{X} = \{x_u : u \in \mathcal{V}\}$. In addition, let $\dim(A)$ to denote the number of rows(or columns) $n$ of the square matrix $A \in \{0, 1\}^{n \times n}$. We describe the full procedure in Algorithm 1. Note that the time complexity of the procedure is $O(N^2)$ (Brisaboa et al., 2009), where $N$ denotes the number of nodes in the graph $G$.

# B  CONSTRUCTING A GRAPH FROM THE $K^2$–TREE

---

**Algorithm 2** Graph $G$ construction

---

**Input**: $K^2$–tree $(\mathcal{T}, \mathcal{X})$ and partitioning factor $K$.
Set $m \leftarrow K^{D_{\mathcal{T}}}$.                                                          ▷ Full adjacency matrix size.
Initialize $A \in \{0,1\}^{m \times m}$ with zeros.
**for** $u \in \mathcal{L}$ **do**                                                                  ▷ For each leaf node with $x_u = 1$.
    $\mathrm{pos}(u) = ((i_{v_1}, j_{v_1}), \ldots, (i_{v_L}, j_{v_L}))$.                        ▷ Position of node $u$.
    $(p_u, q_u) = (\sum_{\ell=1}^{L} K^{L-\ell}(i_{v_\ell} - 1) + 1, \sum_{\ell=1}^{L} K^{L-\ell}(j_{v_\ell} - 1) + 1)$.   ▷ Location of node $u$.
    Set $A_{p_u, q_u} \leftarrow 1$.
**end for**
**Output**: Adjacency matrix $A$.

---

We next describe the algorithm to generate a graph $G = A$ given the $K^2$–tree $(\mathcal{T}, \mathcal{X})$ with tree depth $D_{\mathcal{T}}$. Let $\mathcal{L} \subset \mathcal{V}$ be the set of leaf nodes in $K^2$–tree with node attributes 1. Note that we represent the tree-node position of $u \in \mathcal{V}$ as $\mathrm{pos}(u) = ((i_{v_1}, j_{v_1}), \ldots, (i_{v_L}, j_{v_L}))$ based on a downward path $v_0, v_1, \ldots, v_L$ from the root node $r = v_0$ to the tree-node $u = v_L$. In addition, the location of corresponding submatrix $A^{(u)}$ is denoted as $(p_u, q_u) = (\sum_{\ell=1}^{L} K^{L-\ell}(i_{v_\ell} - 1) + 1, \sum_{\ell=1}^{L} K^{L-\ell}(j_{v_\ell} - 1) + 1)$ in as described in Section 4.1. We describe the full procedure as in Algorithm 2. Note that the time complexity of the procedure is $O(N^2)$, where $N$ denotes the number of nodes in the graph $G$, since it requires querying for each element in the adjacency matrix.

## C GENERALIZING $K^2$–TREE TO ATTRIBUTED GRAPHS

---

**Algorithm 3** Featured $K^2$–tree construction

---

    **Input**: Modified adjacency matrix $A$ and partitioning factor $K$.
1: Initialize the tree $\mathcal{T} \leftarrow (\mathcal{V}, \mathcal{E})$ with $\mathcal{V} = \emptyset, \mathcal{E} = \emptyset$.          ▷ Featured $K^2$–tree.
2: Initialize an empty queue $Q$.          ▷ Candidates to be expanded into child nodes.
3: Set $\mathcal{V} \leftarrow \mathcal{V} \cup \{r\}$, $x_r \leftarrow 1$ and let $A^{(r)} \leftarrow A$. Insert $r$ into the queue $Q$.      ▷ Add root node $r$.
4: **while** $Q \neq \emptyset$ **do**
5:     Pop $u$ from $\mathcal{Q}$.
6:     **if** $x_u = 0$ **then**          ▷ Condition for not expanding the node $u$.
7:         Go to line 4.
8:     **end if**
9:     Update $s \leftarrow \dim(A^{(u)})/K$
10:     **for** $i = 1, \ldots, K$ **do**          ▷ Row-wise indices.
11:         **for** $j = 1, \ldots, K$ **do**          ▷ Column-wise indices.
12:             Set $B_{i,j} \leftarrow A^{(u)}[(i-1)s : is, (j-1)s : js]$.
                 ▷ Operation to obtain $s \times s$ submatrix $B_{i,j}$ of $A^{(u)}$.
13:             **if** $B_{i,j}$ is filled with zeros **then**          ▷ Update tree-node attribute.
14:                 Set $x_v \leftarrow 0$.
15:             **else if** $|B_{i,j}| > 1$ **then**          ▷ Non-leaf tree-nodes with attribute 1.
16:                 Set $x_v \leftarrow 1$.
17:             **else**          ▷ Leaf tree-nodes with node features and edge features.
18:                 Set $x_v \leftarrow B_{i,j}$.          ▷ We treat $1 \times 1$ matrix $B_{i,j}$ as a scalar.
19:             **end if**
20:             **if** $\dim(B_{i,j}) > 1$ **then** $\mathcal{Q} \leftarrow v_{i,j}$.
21:             **end if**
22:         **end for**
23:     **end for**
24:     Set $\mathcal{V} \leftarrow \mathcal{V} \cup \{v_{1,1}, \ldots, v_{K,K}\}$.          ▷ Update tree nodes.
25:     Set $\mathcal{E} \leftarrow \mathcal{E} \cup \{(u, v_{1,1}), \ldots, (u, v_{K,K})\}$.          ▷ Update tree edges.
26: **end while**
    **Output**: Featured $K^2$–tree $(\mathcal{T}, \mathcal{X})$ where $\mathcal{X} = \{x_u : u \in \mathcal{V}\}$.

---

**Algorithm 4** Featured graph $G$ construction

---

1: **Input**: Featured $K^2$–tree $(\mathcal{T}, \mathcal{X})$ and partitioning factor $K$.
2: $m \leftarrow K^{D_\mathcal{T}}$          ▷ Full adjacency matrix size.
3: Initialize $A \in \{0, 1\}^{m \times m}$ with zeros.
4: **for** $u \in \mathcal{L}$ **do**          ▷ For each leaf node with $x_u \neq 0$.
5:     $\text{pos}(u) = ((i_{v_1}, j_{v_1}), \ldots, (i_{v_L}, j_{v_L}))$.          ▷ Position of node $u$.
6:     $(p_u, q_u) = (\sum_{\ell=1}^{L} K^{L-\ell}(i_{v_\ell} - 1) + 1, \sum_{\ell=1}^{L} K^{L-\ell}(j_{v_\ell} - 1) + 1)$.     ▷ Location of node $u$.
7:     Set $A_{p_u, q_u} \leftarrow x_u$.
8: **end for**
9: **Output**: Modified adjacency matrix $A$.

---

In this section, we describe a detailed process to construct a $K^2$–tree for featured graphs with node features and edge features (e.g., molecular graphs), which is described briefly in Section 4.1. We modify the original adjacency matrix by incorporating categorical features into each element, thereby enabling the derivation of the featured $K^2$–tree from the modified adjacency matrix.

**Edge features.** Integrating edge features into the adjacency matrix is straightforward. It can be accomplished by simply replacing the ones with the appropriate categorical edge features.

**Node features.** Integrating node features into the adjacency matrix is more complex than that of edge features since the adjacency matrix only describes the connectivity between node pairs. To address this issue, we assume that all graph nodes possess self-loops, which leads to filling ones to the diagonal elements. Then we replace ones on the diagonal with categorical node features that correspond to the respective node positions.

Let $x_u \in \mathcal{X}$ be the non-binary tree-node attributes that include node features and edge features and $\mathcal{L}$ be the set of leaf nodes in $K^2$–tree with non-zero node attributes. Then we can construct a featured $K^2$–tree with a modified adjacency matrix and construct a graph $G$ from the featured $K^2$–tree as described in Algorithm 3 and Algorithm 4, respectively.

# D    EXPERIMENTAL DETAILS

In this section, we provide the details of the experiments. Note that we chose $k = 2$ in all experiments and provide additional experimental results for $k = 3$ in Appendix G.

## D.1    GENERIC GRAPH GENERATION

Table 4: **Hyperparameters of HGGT in generic graph generation.**

|  | Hyperparameter | Community-small | Planar | Enzymes | Grid |
|---|---|---|---|---|---|
| Transformer | Dim. of feed-forward network | 512 | 512 | 512 | 512 |
|  | Transformer dropout rate | 0.1 | 0 | 0.1 | 0.1 |
|  | # of attention heads | 8 | 8 | 8 | 8 |
|  | # of layers | 3 | 3 | 3 | 3 |
| Train | Batch size | 128 | 32 | 32 | 8 |
|  | # of epochs | 500 | 500 | 500 | 500 |
|  | Dim. of token embedding | 512 | 512 | 512 | 512 |
|  | Gradient clipping norm | 1 | 1 | 1 | 1 |
|  | Input dropout rate | 0 | 0 | 0 | 0 |
|  | Learning rate | $1 \times 10^{-3}$ | $1 \times 10^{-3}$ | $2 \times 10^{-4}$ | $5 \times 10^{-4}$ |

We used the same split with GDSS (Jo et al., 2022) for Community-small, Enzymes, and Grid datasets. Otherwise, we used the same split with SPECTRE (Luo et al., 2022) for the Planar dataset. We fix $k = 2$ and perform the hyperparameter search to choose the best learning rate in $\{0.0001, 0.0002, 0.0005, 0.001\}$ and the best dropout rate in $\{0, 0.1\}$. We select the model with the best MMD with the lowest average of three graph statistics: degree, clustering coefficient, and orbit count. Finally, we provide the hyperparameters used in the experiment in Table 6.

## D.2    MOLECULAR GRAPH GENERATION

Table 5: **Statstics of molecular datasets: QM9 and ZINC250k**.

| Dataset | # of graphs | # of nodes | # of node types | # of edge types |
|---|---|---|---|---|
| QM9 | 133,885 | $1 \leq |V| \leq 9$ | 4 | 3 |
| ZINC250k | 249,455 | $6 \leq |V| \leq 38$ | 9 | 3 |

Table 6: **Hyperparameters of HGGT in molecular graph generation.**

|  | Hyperparameter | QM9 | ZINC250k |
|---|---|---|---|
| Transformer | Dim. of feedforward network | 512 | 512 |
|  | Transformer dropout rate | 0.1 | 0.1 |
|  | # of attention heads | 8 | 8 |
|  | # of layers | 2 | 3 |
| Train | Batch size | 1024 | 256 |
|  | # of epochs | 500 | 500 |
|  | Dim. of token embedding | 512 | 512 |
|  | Gradient clipping norm | 1 | 1 |
|  | Input dropout rate | 0.5 | 0 |
|  | Learning rate | $5 \times 10^{-4}$ | $5 \times 10^{-4}$ |

The statistics of training molecular graphs (i.e., QM9 and ZINC250k datasets) are summarized in Table 5 and we used the same split with GDSS (Jo et al., 2022) for a fair evaluation. We fix $k = 2$ and perform the hyperparameter search to choose the best number of layers in $\{2, 3\}$ and select the model with the best validity. In addition, we provide the hyperparameters used in the experiment in Table 6.

# E IMPLEMENTATION DETAILS

## E.1 COMPUTING RESOURCES

We used PyTorch (Paszke et al., 2019) to implement HGGT and train the Transformer (Vaswani et al., 2017) models on a single GeForce RTX 3090 GPU.

## E.2 MODEL ARCHITECTURE

We describe the architecture of the proposed transformer generator of HGGT in Figure 8. The generator takes a sequential representation of $K^2$–tree as input and generates the output probability of each token as described in Section 4.2. The model consists of a token embedding layer, transformer encoder(s), and multilayer perceptron layer with tree positional encoding.

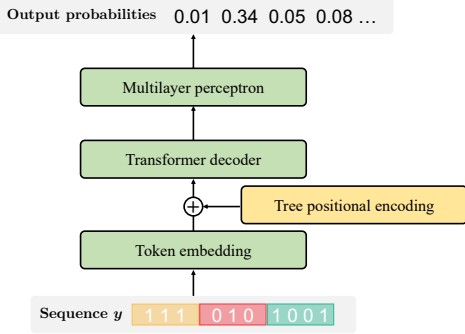

Figure 8: **The architecture of HGGT.**

## E.3 DETAILS FOR BASELINE IMPLEMENTATION

**Generic graph generation.** The baseline results from prior works are as follows. Results for Graph-VAE (Simonovsky & Komodakis, 2018), GraphRNN (You et al., 2018), GNF (Liu et al., 2019), EDP-GNN (Niu et al., 2020), GraphAF (Shi et al., 2020), GraphDF (Luo et al., 2021), and GDSS (Jo et al., 2022) are obtained from GDSS, while the results for GRAN (Liao et al., 2019), SPECTRE (Martinkus et al., 2022), and GDSM (Luo et al., 2022) are derived from their respective paper. Additionally, we reproduced DiGress (Vignac et al., 2022) and GraphGen (Goyal et al., 2020) using their open-source codes. We used original hyperparameters when the original work provided them. DiGress takes more than three days for the Planar, Enzymes, and Grid datasets, so we report the results from fewer epochs after convergence.

**Molecular graph generation.** The baseline results from prior works are as follows. The results for EDP-GNN (Niu et al., 2020), MoFlow (Zang & Wang, 2020), GraphAF (Shi et al., 2020), GraphDF (Luo et al., 2021), GraphEBM (Liu et al., 2021), and GDSS (Jo et al., 2022) are from GDSS, and the GDSM (Luo et al., 2022) result is extracted from the corresponding paper. Moreover, we reproduced DiGress (Vignac et al., 2022) using their open-source codes.

## E.4 DETAILS FOR THE IMPLEMENTATION

We adapted node ordering code from (Diamant et al., 2023), evaluation scheme from (Jo et al., 2022; Martinkus et al., 2022), and NSPDK computation from (Goyal et al., 2020).

## F    GENERATED SAMPLES

In this section, we provide the visualizations of the generated graphs for generic and molecular graph generation.

### F.1    GENERIC GRAPH GENERATION

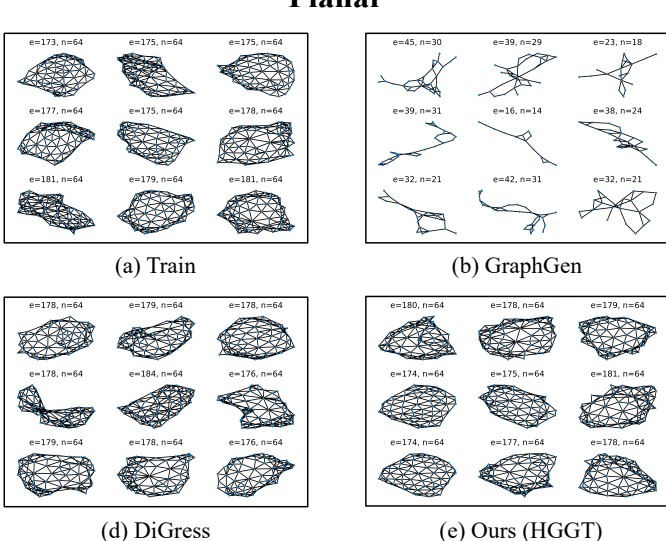

**Community-small**

(a) Train    (b) GraphGen    (c) GDSS

(d) DiGress    (e) Ours (HGGT)

Figure 9: **Visualization of the graphs from the Community-small dataset and the generated graphs.**

**Planar**

(a) Train    (b) GraphGen

(d) DiGress    (e) Ours (HGGT)

Figure 10: **Visualization of the graphs from the Planar dataset and the generated graphs.**

We present visualizations of graphs from the training dataset and generated samples from GraphGen, DiGress, GDSS, and HGGT in Figure 9, Figure 10, Figure 11, and Figure 12. Note that we reproduced GraphGen and DiGress using open-source codes while utilizing the provided checkpoints for GDSS. However, given that the checkpoints provided for GDSS do not include the Planar dataset, we have omitted GDSS samples for this dataset. We additionally give the number of nodes and edges of each graph.

## Enzymes

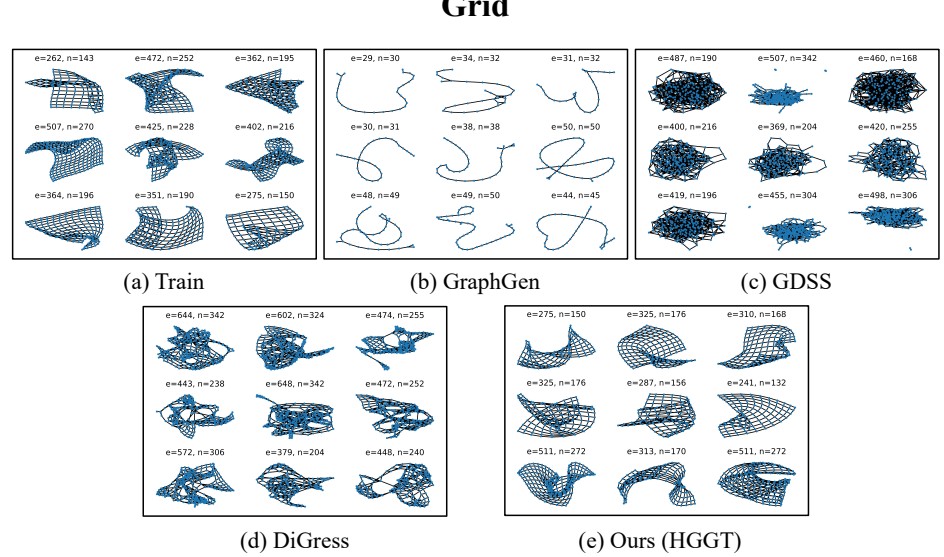

Figure 11: **Visualization of the graphs from the enzymes dataset and the generated graphs.**

## Grid

Figure 12: **Visualization of the graphs from the Grid dataset and the generated graphs.**

## F.2 MOLECULAR GRAPH GENERATION

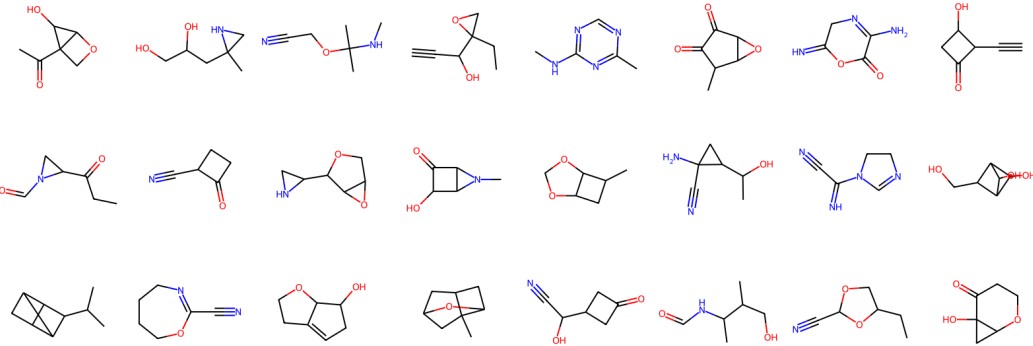

Figure 13: **Visualization of the molecules generated from the QM9 dataset.**

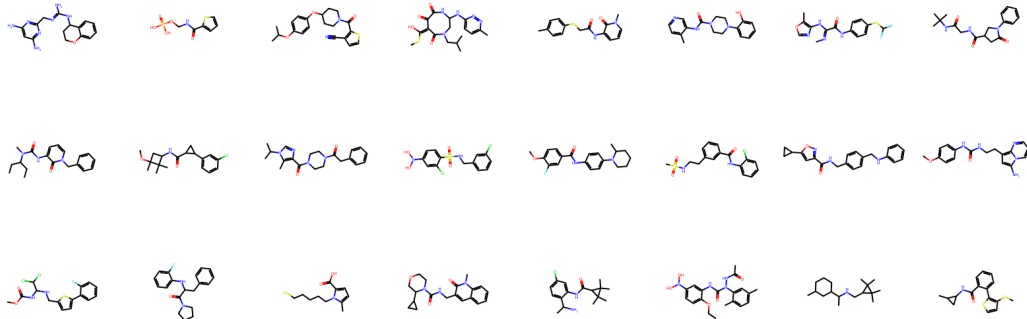

Figure 14: **Visualization of the molecules generated from the ZINC250k dataset.**

We present visualizations of generated molecules from HGGT in Figure 13 and Figure 14. Note that the 24 molecules are non-cherry-picked and randomly sampled.

# G  ADDITIONAL EXPERIMENTAL RESULTS

In this section, we report additional experimental results.

Figure 15: **Generation results of HGGT with $k = 3$.**

| | Community-small | | | Planar | | |
|---|---|---|---|---|---|---|
| | Degree | Cluster. | Orbit | Degree | Cluster. | Orbit |
| $k = 2$ | **0.001** | **0.006** | 0.003 | **0.000** | **0.001** | **0.000** |
| $k = 3$ | 0.007 | 0.050 | **0.001** | 0.001 | 0.003 | **0.000** |

## G.1  GENERIC GRAPH GENERATION

We provide generic graph generation results for $k = 3$. Increasing $k$ decreases the sequence length, while vocabulary size increases to $2^{3^2} + 2^6 = 578$.

We used Community-small and Planar datasets and measured MMD between the test graphs and generated graphs. We perform the same hyperparameter search for a fair evaluation as $k = 2$. The results are in Figure 15. We can observe that HGGT still outperforms the baselines even with different $k$.

## G.2  MOLECULAR GRAPH GENERATION

Table 7: **Additional molecular graph generation performance.**

| Method | QM9 | | | | | | |
|---|---|---|---|---|---|---|---|
| | Frag. ↑ | Intdiv. ↑ | QED ↓ | SA ↓ | SNN ↑ | Scaf. ↑ | Weight ↓ |
| DiGress | 0.9737 | **0.9189** | 0.0015 | **0.0189** | **0.5216** | 0.9063 | **0.1746** |
| HGGT (Ours) | **0.9874** | 0.9150 | **0.0012** | 0.0304 | 0.5156 | **0.9368** | 0.2430 |

| Method | ZINC250k | | | | | | |
|---|---|---|---|---|---|---|---|
| | Frag. ↑ | Intdiv. ↑ | QED ↓ | SA ↓ | SNN ↑ | Scaf. ↑ | Weight ↓ |
| DiGress | 0.7702 | **0.9061** | 0.1284 | 1.9290 | 0.2491 | 0.0001 | 62.9923 |
| HGGT (Ours) | **0.9877** | 0.8644 | **0.0164** | **0.2407** | **0.4383** | **0.5298** | **1.8592** |

We additionally report seven metrics of the generated molecules: (a) fragment similarity (Frag.), which measures the BRICS fragment frequency similarity between generated molecules and test molecules, (b) internal diversity (Intdiv.), which measures the chemical diversity in generated molecules, (c) quantitative estimation of drug-likeness (QED), which measures the drug-likeness similarity between generated molecules and test molecules, (d) synthetic accessibility score (SA), which compares the synthetic accessibility between generated molecules and test molecules, (e) similarity to the nearest neighbor (SNN), an average of Tanimoto similarity between the fingerprint of a generated molecule and test molecule, (f) scaffold similarity (Scaf.), the Bemis-Murcko scaffold frequency similarity between generated molecules and test molecules, and (g) weight, the atom weight similarity between generated molecules and test molecules. The results are in Table 7.

# H  DISCUSSION

## H.1  HIERARCHY OF $K^2$–TREE REPRESENTATION

$K^2$–tree representation is hierarchical as it forms a parent-child hierarchy between nodes. In detail, each node in $K^2$–tree corresponds to a block (i.e., submatrix) in the adjacency matrix. Given a child and its parent node, the child node block is a submatrix of the parent block node, which enables $K^2$–tree to represent a hierarchical structure between the blocks. While this hierarchy may differ from the exact hierarchical community structure, the $K^2$–tree representation still represents a valid hierarchy present in the adjacency matrix. We also note that our $K^2$–tree representation should not be confused with the hierarchical representation learned by graph neural networks with pooling functions (Ying et al., 2018).

Nevertheless, bandwidth minimization algorithms (including C-M node ordering) often induce node orderings that align with underlying clusters. Prior works (Barik et al., 2020; Mueller, 2004) have empirically figured out that bandwidth minimization tends to cluster the points along the diagonal, which leads to a partial capture of the underlying community structure. This also supports our statement that $K^2$–tree representation is hierarchical.

## H.2  COMPARISON WITH PRIOR WORKS ON AUTOREGRESSIVE GRAPH GENERATIVE MODEL

In this section, we compare HGGT to two prior works on autoregressive graph generative models: GraphRNN (You et al., 2018) and GRAN (Liao et al., 2019). The main difference comes from the key idea of HGGT: the ability to capture recurring patterns and the hierarchy of the adjacency matrix. In detail, HGGT maps the recurring patterns in the dataset (large zero-filled block matrices) into a simple object (a zero-valued node in the $K^2$–tree that the model can easily generate. This mapping allows the generative model to focus on learning instance-specific details rather than the generation of the whole pattern that is common across the dataset. In addition, $K^2$–tree representation can represent a valid hierarchy present in the adjacency matrix, as described in Appendix H.1

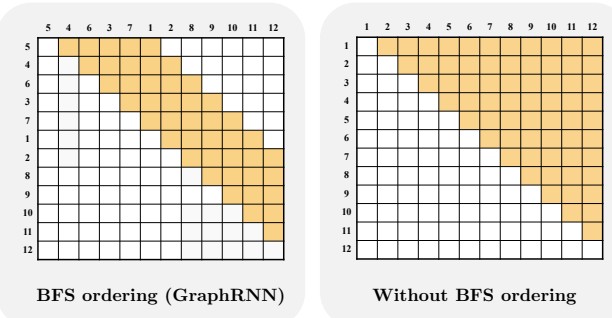

Figure 16: **Reduced representation size of GraphRNN.**

**Comparison to GraphRNN.** Both HGGT and GraphRNN reduce the representation size, which removes the burden of the graph generative models learning long-range dependencies. In detail, HGGT reduces the representation size by leveraging $K^2$–tree, pruning, and tokenization. Otherwise, GraphRNN employs BFS node ordering constraining the upper-corner elements of the adjacency matrix to be consecutively zero as described in Figure 16.

|  | Comm. | Planar | Enzymes | Grid |
|---|---|---|---|---|
| Full matrix | 241.3 | 4096.0 | 1301.6 | 50356.1 |
| GraphRNN | 75.2 | 2007.3 | 234.7 | 4106.3 |
| HGGT (ours) | **30.3** | **211.7** | **67.3** | **419.1** |

Table 8: **Representation size of GraphRNN and HGGT**

| Dataset | Rep. size | $N^2$ |
|---------|-----------|-------|
| Comm. | 48 | 400 |
| Enzymes | 238 | 15625 |
| Planar | 230 | 4696 |
| Grid | 706 | 130321 |

The reduction of HGGT is higher as shown in Table 8 that reports the average size of the representation empirically. Note that the representation size of HGGT and GraphRNN indicates the number of tokens and the number of elements limited by the maximum size of the BFS queue, respectively. While the comparison is not fair due to different vocabulary sizes, one could expect HGGT to suffer less from the long-range dependency problem due to the shorter representation.

**Comparison to GRAN.** The main difference between HGGT and GRAN comes from the different generated representations. HGGT generates $K^2$–tree representation with large zero-filled blocks, which is further summarized into a single node in the $K^2$–tree while GRAN generates the conventional adjacency matrix. Notably, the block of nodes of GRAN is solely used for parallel decoding, which is conceptually irrelevant to the graph representation.

In addition, the concept of block and the decoding process of the blocks differ in both methods. On one hand, the square-shaped HGGT block defines connectivity between a pair of equally-sized node sets. HGGT sequentially specifies these block matrix elements in a hierarchical way, i.e., first specifying whether the whole block matrix is filled with zeros and then specifying its smaller submatrices. On the other hand, the rectangular-shaped GRAN block defines connectivity between a set of newly added nodes and the existing nodes. GRAN can optionally decode the block matrix elements in parallel, speeding up the decoding process at the cost of lower performance.

