# OpenReview forum: "Graph Generation with  $K^2$-trees"
_ICLR.cc/2024/Conference — ICLR 2024 poster_

### Official Review · Reviewer_XLS2 · 2023-10-28

**Soundness:** 3 good
**Presentation:** 3 good
**Contribution:** 2 fair
**Rating:** 5
**Confidence:** 3

**Summary:**

The paper focuses on generating graphs from a target distribution, which is a vital task in several areas like drug discovery and social network analysis. The paper introduces a new framework termed as "Hierarchical Graph Generation with K^2−Tree" (HGGT). This model (1) uses a K^2−Tree representation, which was originally designed for lossless graph compression, enabling a compact graph representation while also capturing the hierarchical structure of the graph. (2) Incorporates pruning, flattening, and tokenization processes in the K^2−Tree representation. (3) Introduces a Transformer-based architecture, optimized for generating sequences by using a specialized tree positional encoding scheme.

**Strengths:**

1. The paper's emphasis on hierarchically capturing graph structures using K^2−Tree is technically sound. Hierarchies are crucial for many real-world graphs, and K^2−Tree, with its inherent structure, naturally offers this advantage.

2. The introduction of pruning, flattening, and tokenization processes aims to achieve a compact representation. This can lead to both storage and computational efficiencies, which are pivotal when dealing with large-scale graph data.

**Weaknesses:**

1. I have doubts about the motivation, which is not strong enough to drive the development of such work.

2. Related work missing, such as [1, 2]

3. The paper doesn't detail the computational resources required, which raises concerns about its practicality for very large graphs.

4. Tokenization can sometimes lead to loss of information, and without details, it's uncertain how this impacts the overall graph representation.

[1] Kong, Lingkai, et al. "Autoregressive Diffusion Model for Graph Generation." (2023).

[2] Chen, Xiaohui, et al. "Efficient and Degree-Guided Graph Generation via Discrete Diffusion Modeling." (2023)

**Questions:**

See weakness, I'd like to raise my score if concern is addressed

---

> ### Author Response · Authors · 2023-11-15
>
> Dear reviewer XLS2,
>
> We sincerely appreciate your comments and efforts in reviewing our paper. We address your question as follows. We also updated our manuscript which is highlighted in $\color{red}{\text{red}}$.
>
> **W1. Motivation is not strong enough.**
>
> To alleviate your concern, we first restate our motivation stated in our introduction (development of compact and hierarchical representation) and then strengthen it by providing the underlying idea (exploiting large zero-filled submatrices) that is more explicit.
>
> The motivation stated in our introduction is the benefits of compact and hierarchical representation for graph generation. Compact representation reduces the complexity of graph generation and simplifies the search space over graphs. Hierarchical representation allows generating the graph in a coarse-to-fine way, which aligns with the hierarchical nature of real-world graphs.
>
> Our idea beneath this motivation is to map the patterns (zero-filled blocks) being repeated across the whole dataset into simple objects (nodes in the $K^2$-tree) that the model can easily generate. This mapping removes the burden of the generative model to learn how to generate large patterns and allows the model to focus on learning instance-specific details.
>
> We also note that our $K^2$-tree being hierarchical allows us to handle various patterns (zero-filled blocks with varying sizes) in a systematic way. Our $K^2$-tree being compact for a dataset implies that the patterns (zero-filled blocks) indeed frequently appear across the whole dataset.
>
> We believe this idea is strong enough to drive our work since similar ideas showed success in different domains in different forms.
> - Most of the successful language models [1] generate a sentence word-by-word instead of character-by-character. Here, the recurring patterns of characters are mapped into word tokens.
> - Several successful molecular generative models [2, 3] also generate molecules fragment-by-fragment. Here, the recurring patterns of atoms and bonds are mapped into fragments.
>
> ---
>
> **W2. Related works are missing, such as GraphARM and degree-guided diffusion.**
>
> Thank you for pointing out the references. We added the references about GraphARM and the degree-guided diffusion model in our updated manuscript.
> - GraphARM (ICML 2023) is a very recent work that improves over the existing diffusion models based on the forward and reverse diffusion process that iteratively masks and unmasks the graph elements in an autoregressive way.
> - Degree-guided diffusion (ICML 2023) is also a very recent work that improves the scalability of the diffusion models for graph generation. It exploits graph sparsity based on adding edges between only a small portion of nodes at each graph construction (reverse diffusion) step.
>
> ---
>
> **W3. The paper doesn't detail the computational resources.**
>
> We used a single GeForce RTX 3090 GPU to train and generate samples with HGGT. This was detailed in Appendix E.1 in our original manuscript. We added the description to our main paper for better visibility in the updated manuscript.
>
> ---
>
> **W4. Tokenization can sometimes lead to loss of information, and without details, it's uncertain how this impacts the overall graph representation.**
>
> We would like to clarify that our tokenization does not lose any information. Our tokenization is a mapping from $K^2$ binary digits to a vocabulary of size 24 where one can perfectly reconstruct any $K^2$-tree after the tokenization process. We systematically designed our tokenization process so that there is no unknown token (<UNK>) that may lead to loss of information as the reviewer mentioned.
>
>
> **References**
>
> [1] Devlin, J., et al., BERT: Pre-training of deep bidirectional transformers for language understanding. Association for Computational Linguistics 2018.
>
> [2] Jin, W., et al., Junction tree variational autoencoder for molecular graph generation. ICML 2018.
>
> [3] Jin, W., et al., Hierarchical generation of molecular graphs using structural motifs. ICML 2020.

---

> > ### Author Response · Authors · 2023-11-23
> >
> > Dear reviewer XLS2,
> >
> > Thank you for your time and efforts in reviewing this paper. Since the discussion phase is close to the end, we would like to inquire if our responses have addressed your concerns.
> >
> > We are especially curious since you indicated that your assessment of our paper was borderline and would like to change the score if your concerns are addressed. We wonder if our response has successfully addressed the issues.
> >
> > We remain fully committed to addressing any questions you may have by the end of the discussion phase.

---

### Official Review · Reviewer_K9N3 · 2023-10-29

**Soundness:** 3 good
**Presentation:** 4 excellent
**Contribution:** 3 good
**Rating:** 5
**Confidence:** 4

**Summary:**

This paper presents a new graph generative model which capitalizes on the $K^2-$tree representation, a  representation of graphs that is more compact than the adjacency matrix. The $K^2-$tree representation is transformed into a sequence and then a Transformer-based architecture is employed which predicts one token at a time based on the previously generated sequence. The Transformer is also equipped with positional encodings that take into account the structure of the tree. The proposed model is trained on synthetic and real-world datasets. The results indicate that in most cases, the generated graphs better preserve graph properties than the baselines.

**Strengths:**

- The proposed model shows strong empirical performance over previous baselines on both synthetic and real-world datasets. Thus, HGGT could be a useful addition to the list of graph generative models.

- The $K^2-$tree representation seems interesting and the proposed model has some novelty. Even though there are previous works that have proposed autoregressive models for graph generation, in my view the main components of HGGT are different from those of previous works.

- The model supports node and edge features, while the results reported in Figure 7 suggest that HGGT is much more efficient than competing models.

**Weaknesses:**

- The paper claims that the employed representation is hierarchical, however, I do not fully agree with this claim. In case a hierarchical community structure is present in the graph, a hierarchical representation is supposed to capture this community structure. However, the proposed $K^2-$tree representation would not necessarily capture this (since it depends on the node ordering). On the other hand, the hierarchical clustering algorithm would produce a proper hierarchical representation. I thus think that this claim needs rephrasing to avoid misunderstanding.

- The proposed model is conceptually similar to GRAN [1] which sequentially generates blocks of nodes and associated edges. A detailed discussion of how HGGT differs from GRAN is missing from the paper.

- One of my main concerns with this work is that it is not clearly explained in the paper why the proposed model significantly outperforms the baselines. This is not the first autoregressive model for graph generation, and previous models also came up with different schemes to reduce the time and space complexity (such as BFS ordering and generation of blocks of the adjacency matrix in [2] and [1], respectively). Thus, I would not expect such a significant difference in performance between HGGT and those previous models. I would like the authors to comment on this.

- In Table 2, we can observe that the novelty of the generated molecules is low compared to those of the baselines (mainly on QM9). I would expect the authors to provide some explanation or intuitions about why the proposed model fails to produce novel graphs.

- In section 5.2, it is mentioned that "Each metric is measured between the 10,000 generated samples and the test set". I do not think that this is actually true. If I am not wrong the validity and the uniqueness have nothing to do with the samples of the test set. Furthermore, the Frechet ChemNet Distance and the novelty are commonly computed by comparing the generated samples against those of the training set and not those of the test set.

[1] Liao, R., Li, Y., Song, Y., Wang, S., Hamilton, W. L., Duvenaud, D., Urtasun, R., & Zemel, R. "Efficient graph generation with graph recurrent attention networks". In Proceedings of the 33rd International Conference on Neural Information Processing Systems, pp. 4255-4265, 2019.\
[2] You, J., Ying, R., Ren, X., Hamilton, W., & Leskovec, J. "Graphrnn: Generating realistic graphs with deep auto-regressive models". Proceedings of the 35th International Conference on Machine Learning, pp. 5708-5717, 2018.

**Questions:**

In p.5, why $K^2$ elements are not enough and $K(K + 1)/2$ more elements are added, thus increasing the vocabulary size for each token?

---

> ### Author Response · Authors · 2023-11-15
>
> Dear reviewer K9N3,
>
> We sincerely appreciate your comments and efforts in reviewing our paper. We address your question as follows. We also updated our manuscript which is highlighted in $\color{red}{\text{red}}$.
>
> ---
> **W1. The claim "capturing the hierarchical graph structure" and the description "hierarchical representation" need rephrasing.**
>
> Thank you for the insightful comment. We agree that our claims on "capturing the hierarchical graph structure" can be ambiguous. In our updated manuscript, we added a discussion on how our HGGT does not necessarily capture the exact hierarchical community structure in Appendix H.
>
> However, we believe it is still appropriate to describe our $K^2$-tree as a hierarchical representation, hence choose to precisely specify the terminology instead of entirely rephrasing it. Our precise description is incorporated in the updated manuscript and explained in what follows.
>
> Nodes in the $K^2$-tree form a parent-child hierarchy between the nodes. To be specific, each node in our $K^2$-tree corresponds to a block (i.e., submatrix) in the adjacency matrix. Given a child and its parent node, the child node block is a submatrix of the parent node block, hence the tree also represents a hierarchical structure between the blocks. While this hierarchy may differ from the hierarchical community structure, the $K^2$-tree representation still represents a valid hierarchy present in the adjacency matrix.
>
> Finally, we also point out prior works [1,2] that imply how the Cuthill–McKee node ordering (used in our HGGT) partially captures the underlying community structure. These prior works additionally support our statement that our $K^2$-tree representation is a hierarchical representation. This is incorporated in Appendix H of our updated manuscript.
>
> ---
>
> **W2. A detailed discussion of how HGGT differs from GRAN is missing.**
>
> Thank you for pointing this out. We compare our HGGT with GRAN in the updated manuscript (Appendix H) and in what follows.
>
> First, the main conceptual difference is in the representation generated by HGGT and GRAN.
> - HGGT generates $K^2$-tree representation for graphs with large zero-filled blocks. This is based on summarizing the large zero-filled blocks into a single node in the $K^2$-tree.
> - GRAN generates the conventional adjacency matrix representation. The *block matrices are conceptually irrelevant for the graph representation* and instead used for parallel decoding.
>
> Next, as the reviewer mentioned, both HGGT and GRAN use block-wise decoding. However, the decoding processes of the block matrix elements are different.
> - HGGT sequentially specifies the block matrix elements in a hierarchical way. It first specifies whether the whole block matrix is filled with zeros at the upper level of the $K^2$-tree, then proceeds to specify smaller submatrices in the lower levels of the $K^2$-tree.
> - GRAN can optionally decode the block matrix elements in parallel. All the elements in the block matrices are generated at once to speed up the decoding process (at the cost of slightly lower performance).
>
> Finally, the semantics and shapes of the block matrix in HGGT and GRAN are also different.
> - The square-shaped HGGT block defines connectivity between a pair of equally-sized node sets.
> - The rectangular-shaped GRAN block defines connectivity between a set of newly added nodes and the existing nodes at each decoding step.
>
> This is incorporated in Appendix H of our updated manuscript.

---

> ### Author Response · Authors · 2023-11-15
>
> **W3. It is not clearly explained in the paper why the proposed model significantly outperforms the baselines. Previous models also came up with different schemes to reduce the time and space complexity.**
>
> Our main intuition on why HGGT outperforms the baselines is that only our HGGT maps the recurring patterns in the dataset (large zero-filled block matrices) into a simple object (a zero-valued node in the $K^2$-tree) that the model can easily generate. This mapping allows the generative model to focus on learning instance-specific details rather than the generation of the whole pattern that is common across the dataset. The baseline space and memory reduction techniques do not yield this benefit, e.g., GRAN and GraphRNN propose parallel decoding and constraining the search space, respectively.
>
> In what follows, we make the one-to-one comparison between our scheme and the existing space and memory reduction schemes. Note that we do *not* claim HGGT to strictly improve over the baselines in a conceptual way. In fact, it is hard to make an apple-to-apple comparison between the reduction techniques since the ideas are orthogonal, e.g., the $K^2$-tree can be generated via parallel decoding (like GRAN) and made smaller by constraining the adjacency matrix (like GraphRNN).
>
> **Comparison with GRAN.** As already discussed in **W2**, GRAN employs block-wise parallel decoding to reduce the number of decoding steps for generating the whole adjacency matrix. The parallel decoding does not reduce representation size and the search space, hence lowering the performance for faster generation.
>
> **Comparison with GraphRNN.** While GraphRNN also reduces the representation size (and alleviates the long-range dependency problem), our reduction is higher.
>
> To be specific, GraphRNN limits the search space for adjacency matrices by constraining the "upper-corner" elements of the adjacency matrix to be consecutively zero, based on the maximum size of the BFS queue. See Figure 15 in Appendix H for an illustration.
>
> In comparison, while our HGGT does not put constraints on the graphs being generated, both HGGT and GraphRNN reduce the representation size; this removes the burden of the graph generative models learning long-range dependencies. Empirically, our reduction is higher as shown in the below table that reports the average length of representations for the Community, Planar, Enzymes, and Grid datasets. Note that the representation size of HGGT and GraphRNN indicates the number of tokens and the number of elements limited by the maximum size of the BFS queue, respectively. While the comparison is not fair due to different vocabulary sizes, one could expect HGGT to suffer less from the long-range dependency problem due to the shorter representation.
>
>
> |             | Comm. | Planar | Enzymes |    Grid |
> | ----------- | -----:| ------:| -------:| -------:|
> | Full matrix | 241.3 | 4096.0 |  1301.6 | 50356.1 |
> | GraphRNN    |  75.2 | 2007.3 |   234.7 |  4106.3 |
> | HGGT (Ours) |  30.3 |  211.7 |    67.3 |   419.1 |
>
> This is incorporated in Appendix H of our updated manuscript.
>
> ---
>
> **W4. Why is the result on QM9 not very good, especially for Novelty? Does it indicate the inferiority of HGGT on novel graph generation?**
>
> We first note that graph generative models that faithfully learn the training dataset are more likely to assign high likelihoods on the training dataset, hence the trade-off between the quality and the novelty of graph generation is inevitable. This trade-off is especially significant for the QM9 dataset since it consists of a large number of molecules (134k) despite the small search space (molecules with up to only nine heavy atoms). Indeed, no baseline has successfully avoided this trade-off in the QM9 dataset, e.g., Digress achieves a good FCD at the cost of low novelty in Table 2. We chose to achieve high scores for faithfully learning the underlying distribution (FCD and NSPDK) since we consider them to be more meaningful as evidence for high-quality molecule generation.
>
> Instead of the QM9 results, we would like to put more emphasis on the ZINC250k dataset with a larger search space and higher relevance to real-world applications (drug discovery). In the corresponding result, our HGGT achieves much better FCD and NSPDK metrics compared to the baselines, at the cost of a very small decrease in uniqueness and novelty. We also note that one can compensate for the non-unique and non-novel molecules by filtering them out and generating new molecules.

---

> ### Author Response · Authors · 2023-11-15
>
> **W5. There exists some mis-explanation for molecular evaluation metrics.**
>
> Thank you for pointing this out. Indeed, the validity and the uniqueness are computed with the generated samples themselves and the novelty is computed by comparing with the training set. We fixed our description in Section 5.2. of our updated manuscript.
>
> Otherwise, following the MOSES benchmark [3], FCD and NSPDK are measured between the 10,000 generated samples and the test set. It is also notable that we followed other prior works on graph generative models [4, 5] that reported FCD between generated samples and test sets.
>
> ---
>
> **Q1. Why do you choose the $2^{K^2} + 2^{K(K+1)/2}$ vocabulary instead of $2^{K^2}$ which leads to the larger vocabulary size?**
>
> Our main insight is that the semantics of the diagonal blocks are different from non-diagonal blocks in the $K^2$-tree, as described in the fourth paragraph of Section 4.1. To incorporate this idea, we assign different vocabulary for the diagonal (size $2^{K(K+1)/2}$) and non-diagonal blocks (size $2^{K^{2}}$), resulting in a larger vocabulary size. In practice, we choose $K=2$ and the increase in vocabulary size ($2^{K(K+1)/2}=8$) is not significant.
>
>
> **References**
>
> [1] Barik, R., et al., Vertex reordering for real-world graphs and applications: An empirical evaluation. IISWC 2020.
>
> [2] Mueller, C., et al., Sparse matrix reordering algorithms for cluster identification. Machine Learning in Bioinformatics 2004.
>
> [3] Polykovskiy, D., et al., Molecular sets (MOSES): a benchmarking platform for molecular generation models. Frontiers in pharmacology 2020.
>
> [4] Jo, J., et al., Score-based generative modeling of graphs via the system of stochastic differential equations. ICML 2022.
>
> [5] Luo, T., et al., Fast graph generative model via spectral diffusion. arXiv preprint 2022.

---

> > ### Comment · Reviewer_K9N3 · 2023-11-21
> > **Official Comment by Reviewer K9N3**
> >
> > I thank the authors for their response. After reading the response, I think some of my concerns are not well addressed.
> >
> > I still feel that the use of the term "hierarchical representation" is somewhat misleading. Even though, this is actually a hierarchical representation, this term has a different meaning in this context. For example, there is a large body of works that propose models which learn hierarchical representations such as [1] and [2], and those representations capture the community structure of the input graph. I feel that the discussion added in the appendix does not address this issue.
> >
> > The explanation given in my question on why the proposed model significantly outperforms the baselines makes sense in general. However, it assumes that the imposed node ordering is optimal. How likely is this to happen? Does the employed algorithm provide any guarantees? Also $K$ is set equal to small values, and the emerging trees are not that small in practice. I wonder whether the increase in performance provided by HGGT is due to the Transformer (which is much more complex than the RNNs employed in previous works).
> >
> > In case of very dense graphs, how would the authors expect the HGGT model to perform? Would it still be more efficient than the baselines?
> >
> > I am not fully convinced by the claim that the search space is small on the QM9 dataset. There exist models that achieve very high values of novelty. Moreover, in some applications, generative models are expected to produce only novel samples. How did the authors choose to achieve high scores for FCD and NSPDK? Is there some hyperparameter that allows the model to put more emphasis on them than on validity? More importantly, why do the authors consider FCD and NSPDK to be more meaningful as evidence for high-quality molecule generation? Can the authors provide some reference to back up their claim?
> >
> > [1] Ying, Z., You, J., Morris, C., Ren, X., Hamilton, W., & Leskovec, J. (2018). Hierarchical graph representation learning with differentiable pooling. Advances in Neural Information Processing Systems.\
> > [2] Cangea, C., Veličković, P., Jovanović, N., Kipf, T., & Liò, P. (2018). Towards sparse hierarchical graph classifiers. arXiv:1811.01287.

---

> > > ### Author Response · Authors · 2023-11-22
> > >
> > > Dear reviewer K9N3,
> > >
> > > Thank you for checking our response. We do not take your time and effort for granted. We further respond to your concerns one by one.
> > >
> > > **I still feel that the use of the term "hierarchical representation" is somewhat misleading. Even though this is actually a hierarchical representation, this term has a different meaning in this context. For example, there is a large body of work that proposes models that learn hierarchical representations such as [1] and [2], and those representations capture the community structure of the input graph.**
> > >
> > > Regrettably, we are unable to fully understand the following comment: *"Even though this is actually a hierarchical representation, this term has a different meaning in this context."* We would be happy to further incorporate your comments if your concern is not fully resolved and if you could be more specific about your concerns.
> > >
> > > Nonetheless, to resolve your concern without uncertainty, *we entirely removed the term "hierarchical representation" from our revised manuscript*. We instead say that our model generates graphs in a hierarchical procedure and it captures the hierarchical structure in the adjacency matrix. We hope this alleviates your concern about the term "hierarchical representation."
> > >
> > > Additionally, we added the suggested reference [1] in Appendix H.1 of our revised manuscript to clarify that our term "hierarchy" does not exactly match the hierarchical community structure as the reference [1]. However, to our best knowledge, the reference [2] also uses the term "hierarchical" despite not necessarily capturing the hierarchical community structure, similar to our original manuscript. The reference [2] proposes to pool the graphs via dropping nodes with respect to task-dependent learnable scores and the result does not necessarily coincide with the community structure.
> > >
> > >
> > > **The explanation given in my question on why the proposed model significantly outperforms the baselines makes sense in general. However, it assumes that the imposed node ordering is optimal. How likely is this to happen? Does the employed algorithm provide any guarantees? Also, $K$ is set equal to small values, and the emerging trees are not that small in practice.**
> > >
> > > As we show in Figure 7 of our original manuscript, the $K^2$-trees emerging from our C-M node ordering are actually quite small in practice even with small $K=2$ (due to the exponential growth of zero-filled blocks). Even changing node ordering to BFS and DFS orderings keeps tree size smaller than the original adjacency matrix. This means that the assumption used for our explanation generally holds in practice. We defer the establishment of a guarantee for future work since our focus is on practical improvements.
> > >
> > > **I wonder whether the increase in performance provided by HGGT is due to the Transformer (which is much more complex than the RNNs employed in previous works).**
> > >
> > > While it is hard to compare the model architectures in an apple-to-apple way, we note that GRAN also employs the attention layer similar to our architecture. Furthermore, GDSS and Digress are also based on the Transformer architecture, i.e., they utilize multi-head attention.
> > >
> > > To further alleviate your concern, we provide empirical results on applying a Transformer to the flattened adjacency matrix (similar to GraphRNN and GRAN). We can observe that simply applying a Transformer to any representation of graphs does not enhance the graph generation performance. This also validates the effectiveness of our $K^2$-tree representation.
> > >
> > > |                         |       | Com-small |       |       | Planar |       |
> > > |:-----------------------:|:-----:|:---------:|:-----:|:-----:|:------:|:-----:|
> > > |                         | Deg.  |   Clus.   | Orb.  | Deg.  | Clus.  | Orb.  |
> > > |  Flattened adj. (LSTM)  | 0.189 |   0.052   | 0.043 |N.A.  |  N.A.   |  N.A.  |
> > > | Flattened adj. (Trans.) | 0.283 |   0.286   | 0.076 |  OOM |  OOM   |  OOM  |
> > > |       HGGT (Ours)       | 0.001 |   0.006   | 0.003 | 0.000 | 0.001  | 0.000 |
> > >
> > > It is notable that we report ablation studies in Table 3 of our original manuscript to validate the effectiveness of each component of HGGT such as tokenization, tree positional encoding, and pruning. This additionally indicates how our performance is not just obtained from simply applying the Transformer architecture.
> > >
> > >
> > > **In the case of very dense graphs, how would the authors expect the HGGT model to perform?**
> > >
> > > For extreme cases like fully connected graphs, we expect the $K^2$-tree to perform worse since its size may become larger than the original adjacency matrix. However, this rarely happens for real-world graphs, e.g., for the problems considered in our work, as shown in the lower left Table of Figure 7.

---

> > > > ### Author Response · Authors · 2023-11-22
> > > >
> > > > **I am not fully convinced by the claim that the search space is small on the QM9 dataset.**
> > > >
> > > > The search space of QM9 is smaller than ZINC250k since it consists of molecules with a smaller number of atoms and atom types, e.g., QM9 consists of 133,885 molecules with up to 9 heavy atoms of 4 types while ZINC250k consists of 249,455 molecules with up to 38 heavy atoms of 9 types. In detail, QM9 is an enumeration of possible configurations of a given atom combinations, while ZINC250k consists of real-world drug-like molecules (hence more practically relevant). Therefore, the search space of QM9 is smaller and simpler than ZINC250k.
> > > >
> > > > **There exist models that achieve very high values of novelty. Moreover, in some applications, generative models are expected to produce only novel samples.**
> > > >
> > > > As previously mentioned in our response, the models with high novelty have worse FCD and NSPDK scores (e.g., baselines except for DiGress in Table 2) while the model with low novelty showed better FCD and NSPDK (e.g., DiGress in Table 2). Moreover, for practical applications like ZINC250k, our HGGT achieves high novelty.
> > > >
> > > > If there exist important applications where only novel samples are needed, one can filter out the duplicated samples by accessing the training dataset. Note this procedure is required for any of the considered baselines since none of the baselines can "only" generate novel samples.
> > > >
> > > > **How did the authors choose to achieve high scores for FCD and NSPDK? Is there some hyperparameter that allows the model to put more emphasis on them than on validity?**
> > > >
> > > > We clarify that we do not put more emphasis on FCD and NSPDK than on validity in hyperparameter tuning. We chose the hyperparameters based on validity as explained in Appendix D.2., which can be cheaply evaluated without accessing the test set. This consequently resulted in high FCD and NSPDK scores.
> > > >
> > > > **More importantly, why do the authors consider FCD and NSPDK to be more meaningful as evidence for high-quality molecule generation? Can the authors provide some references to back up their claim?**
> > > >
> > > > We mainly refer to DiGress [3] and MOSES [4] to back up our claim on considering novelty in the QM9 dataset to be less meaningful for the QM9 dataset. In the DiGress paper [3], the authors explained the reason for not reporting novelty as: *"QM9 is an exhaustive enumeration of the small molecules that satisfy a given set of constraints, generating molecules outside this set is not necessarily a good sign that the network has correctly captured the data distribution."*  Furthermore, MOSES [4] states *"For a general evaluation, we suggest using FCD/Test metric that captures multiple aspects of other metrics in a single number."* Our reasoning (explained in our previous response) coincides with the prior works.
> > > >
> > > > **References**
> > > >
> > > > [1] Ying, Z., et al., Hierarchical graph representation learning with differentiable pooling. NIPS 2018.
> > > >
> > > > [2] Cangea, C., et al., Towards sparse hierarchical graph classifiers. arXiv 2018.
> > > >
> > > > [3] Vignac, C., et al. DiGress: Discrete denoising diffusion for graph generation. ICLR 2023.
> > > >
> > > > [4] Polykovskiy, D., et al., Molecular sets (MOSES): a benchmarking platform for molecular generation models. Frontiers in pharmacology 2020.

---

### Official Review · Reviewer_JeGt · 2023-10-30

**Soundness:** 3 good
**Presentation:** 3 good
**Contribution:** 3 good
**Rating:** 8
**Confidence:** 4

**Summary:**

In this paper, the authors propose a new graph generative model Hierarchical Graph Generation with $K^2$–Tree (HGGT). $K^2$-tree is a lossless graph representation and the authors compress it by pruning, flattening and tokenizing operations such that it fits to Transformer with $K^2$-tree positional encoding for graph generation. The effectiveness and efficiency of HGGT are evaluated on six datasets.

**Strengths:**

(1) The approach of combining $K^2$-tree compressed representation with Transformer is new.

(2) The performance of HGGT is superior to the SOTA baselines on most datasets.

**Weaknesses:**

(1) The performance of HGGT (Table 2) is not so satisfactory for molecular graph generation which is probably the most important application of this graph generative model.

(2) It lacks the worst case time complexity analysis for the algorithms.

**Questions:**

(1) Why is the performance of HGGT on molecular datasets not so good as that on the generic graph datasets? It seems that HGGT achieves the worst score on three metrics of the two molecular benchmarks (Uniqueness on QM9 and Novelty on both).

(1) What are the time complexities of Algorithms 1-4 and HGGT?

(2) Is the $K^2$-representation still lossless after pruning, flattening and tokenization? I guess yes, but is there a simple proof for this?

---

> ### Author Response · Authors · 2023-11-15
>
> Dear reviewer JeGt,
>
> We sincerely appreciate your comments and efforts in reviewing our paper. We address your question as follows. We also updated our manuscript which are highlighted in $\color{red}{\text{red}}$.
>
>
> ---
>
> **W1/Q1. The performance of HGGT (Table 2) is not so satisfactory for molecular graph generation. Why is the performance not so good as that on the generic graph datasets?**
>
> We believe our molecular graph generation performance to be satisfactory since it achieves better NSPDK, FCD, QED, and SA as reported in Section 5.2 and Appendix G.2. However, we resonate with your concern since the HGGT scores low novelty and uniqueness for the QM9 dataset, which may yield the impression that our scores are not so satisfactory. To alleviate your concern, we explain why (a) low novelty is a natural consequence of faithfully learning the molecular distribution and (b) one should put more emphasis on the ZINC250k dataset when interpreting our results.
>
> First, we note that the models make a tradeoff between the quality (e.g,. NSPDK and FCD) and novelty of the generated graph since the graph generative models that faithfully learn the distribution put a high likelihood on the training dataset. In particular, the tradeoff is more significant in QM9 due to the large dataset size (134k) compared to the relatively small search space (molecular graphs with only up to nine heavy atoms). It is noteworthy that no existing baseline has successfully avoided this trade-off on the QM9 dataset.
>
> Instead of the QM9 results, we would like to put more emphasis on our results on the ZINC250k with a larger search space and higher relevance to the real-world application, i.e., drug discovery. Here, one can observe our HGGT to achieve better FCD and NSPDK metrics compared to the baselines, while trading off for a very small decrease in uniqueness and novelty.
>
> ---
>
> **Q2/W2. What is the time complexity of $K^2$-tree construction and graph construction from $K^2$-tree?**
>
> The time complexity of $K^2$-tree construction from the adjacency matrix is $O(N^2)$ where $N$ is the number of vertices in the graph, as stated in the original paper [1] that proposed the $K^2$-tree for graph compression. The adjacency matrix construction algorithm also takes $O(N^2)$ time complexity since it requires querying for each element in the adjacency matrix. This is incorporated in Apepndix A and B of our updated manuscript.
>
> ---
>
> **Q3. Is the $K^2$-representation still lossless after pruning, flattening and tokenization?**
>
> Yes. Given a $K^2$-tree, the pruning process iteratively removes each $(i, j)$-th child node associated with a redundant submatrix, i.e., a submatrix consisting of matrix elements positioned above the matrix diagonal. Each removal can be inverted by duplicating the $(j, i)$-th child node as a new $(i, j)$-th child node.
>
> To implement the lossless reconstruction in a simple way, one could apply Algorithm 2 (in Appendix B) to the pruned $K^2$-tree to obtain an incomplete adjacency matrix $\hat{A}$ with zero entries above the diagonal. Then one can recover the full matrix by duplicating the entries below the diagonal, i.e., set the adjacency matrix by $A = \hat{A}+\hat{A}^{\top}$.
>
> In addition, the flattening process is lossless since it flattens the tree with breadth-first traversal. The un-flattening process is defined by adding all the child nodes for each token to a uniquely defined parent node, i.e., the oldest node with label one and without a child node. This process is lossless since it uniquely maps a token to each sequence to a unique $K^2$-tree structure.
>
> To further verify our claim, we conducted experiments on the reconstruction of the original adjacency matrices from the pruned $K^2$-trees, and the results showed a 100% reconstruction rate. This finding confirms that pruned $K^2$-tree is a lossless compression method.
>
> **References**
>
> [1] Brisaboa et al., $K^2$-Trees for Compact Web Graph Representation. International Symposium on String Processing and Information Retrieval 2009.

---

### Official Review · Reviewer_Nzff · 2023-11-06

**Soundness:** 3 good
**Presentation:** 4 excellent
**Contribution:** 3 good
**Rating:** 8
**Confidence:** 3

**Summary:**

The paper proposes a novel algorithm to generate graphs based upon $K^2$-tree representation. One of the positive sides of $K^2$-tree representation lays in the fact that it ensures the compactness of the obtained representation without losing the hierarchical information from the nodes and edges in the original graph. After having described how $K^2$-tree representation works, the authors outline the generation algorithm built upon it. Specifically, the algorithm prunes redundant nodes from the representation (e.g., given its symmetrical nature); the, it flattens and tokenizes the pruned $K^2$-tree; finally, it exploits a Transformer architecture to generate the new graph through positional encoding. Results on various graph learning tasks and domains against other state-of-the-art graph generation solutions outline the efficacy of the proposed approach. The evaluation is complemented through an extensive ablation study which further validates the goodness of the algorithm.

**Strengths:**

+ The paper is well-written and easy-to-follow.
+ The proposed algorithm is simple but effective.
+ The proposed algorithm is also able to generate featured graphs (e.g., molecular structures which come with features on graph edges).
+ The experimental analysis is extensive and supports the efficacy of the proposed solution.
+ The code is released at review time.

**Weaknesses:**

- To the best of my knowledge, I cannot see any specific weakness.

**Questions:**

* Could it be possible to adopt the proposed graph generation algorithm to create graphs with specific topological properties (e.g., node degree or clustering coefficient)?

**After the rebuttal.** The rebuttal answered all questions.

---

> ### Author Response · Authors · 2023-11-15
>
> Dear reviewer Nzff,
>
> We sincerely appreciate your comments and efforts in reviewing our paper. We address your question as follows. We also updated our manuscript which is highlighted in $\color{red}{\text{red}}$.
>
> ---
>
> **Q1. Could it be possible to adopt the proposed graph generation algorithm to create graphs with specific topological properties (e.g., node degree or clustering coefficient)?**
>
> Thank you for the suggestion. Yes, one can train a conditional generative model (parameterized by a Transformer) on our representation with the specific topological properties as conditions. One could even extend our framework to controllable generation [1], which would be an interesting avenue for our future research.
>
> **References**
>
> [1] Keskar, N. S., et al., CTRL: A conditional transformer language model for controllable generation. arXiv preprint 2019.

---

> > ### Comment · Reviewer_Nzff · 2023-11-20
> >
> > Dear Authors,
> >
> > thank you for your rebuttal and thank you for answering to my question/suggestion.

---

> > > ### Author Response · Authors · 2023-11-22
> > >
> > > Thank you. Your insightful comments suggested an interesting venue for future work. Once again, many thanks!

---

### Author Response · Authors · 2023-11-21
**General response**

Dear reviewers (**Nzff**, **JeGt**, **K9N3**, and **XLS2**) and area chairs,

We sincerely appreciate your valuable time and effort spent reviewing our manuscript. Since the rebuttal phase is near the end, **we kindly request the reviewers to check our response**. We are happy to discuss any further concerns raised by the reviewers. In what follows, we summarize our response.

Overall, we are encouraged by how the reviewers found our work appealing due to being sound (**XLS2**), novel idea (**Nzff**, **K9N3**), great empirical results (**JeGt**, **Nzff**, **K9N3**), and clear presentation (**Nzff**, **K9N3**).

We also point out important concerns raised by the reviewers, which we have responded to alleviate concerns.

- Reviewers **K9N3** and **XLS2** questioned the motivation of HGGT and why it outperforms the autoregressive baselines. We reinforced our motivation by explaining how HGGT captures recurring patterns in the dataset. We also added a one-to-one comparison with the autoregressive baselines.
- Reviewers **JeGt** and **K9N3** questioned the reason for the low novelty of molecular graph generation. We explained that HGGT achieves better quality (NSPDK and FCD) which is a trade-off of the low novelty, as the model that faithfully learns the distribution puts a high likelihood on the training dataset. In addition, we emphasized that HGGT achieves better results on the ZINC250k dataset, which has a larger search space and includes real-world molecules.
- Reviewers **JeGt** and **XLS2** asked whether the $K^2$-tree representation is lossless after pruning, flattening, and tokenization processes. We answered that each pruning, flattening, and tokenization step is invertible, which verifies the losslessness of our representation. We also conducted additional experiments to verify the reconstruction of the adjacency matrix from our representation, which showed a 100% reconstruction rate.


Finally, we summarize the revisions of our updated manuscript.

- Discussion on the hierarchy of $K^2$-tree representation
- Discussion on the comparison to the prior works on autoregressive graph generative models
- Explanation of the complexity of the construction of $K^2$-tree
- Incorporation of all the editorial comments

We hope that our response and the discussion phase help resolve these issues.


Thank you very much,

Authors

---

### Meta-Review · Area_Chair_gWjH · 2023-12-04

**Metareview:**

This paper gives a new method to generate graphs based on K2 trees.  The reviewers all generally thought the paper introduced novel ideas. The main strengths are the conceptual idea and extensive experimental evaluation.  Some reviewers had concerns about motivation as well as intuitive explanations for empirical results.  I believe these concerns are handled well in the paper and in the responses by the authors.

**Justification For Why Not Higher Score:**

The results are not of large broad interest, but this is a solid paper.

**Justification For Why Not Lower Score:**

The paper makes it about the threshold for acceptance because the results are interesting and the major concerns by reviewers were already addressed in the paper or require minor modification.

---

### Decision · Program_Chairs · 2024-01-16

Accept (poster)